# PROCESS SUPERVISION-GUIDED POLICY OPTIMIZATION FOR CODE GENERATION

## ABSTRACT

Reinforcement learning (RL) with unit test feedback has enhanced large language models' (LLMs) code generation, but relies on sparse rewards provided only after complete code evaluation, limiting learning efficiency and incremental improvements. When generated code fails all unit tests, no learning signal is received, hindering progress on complex tasks. To address this, we propose a Process Reward Model (PRM) that delivers dense, line-level feedback on code correctness during generation, mimicking human code refinement and providing immediate guidance. We explore various strategies for training PRMs and integrating them into the RL framework, finding that using PRMs both as dense rewards and for value function initialization significantly boosts performance. Our approach increases our in-house LLM's pass rate from 28.2% to 29.8% on LiveCodeBench and from 31.8% to 35.8% on our internal benchmark. Our experimental results highlight the effectiveness of PRMs in enhancing RL-driven code generation, especially for long-horizon scenarios.

## 1 INTRODUCTION

The rapid advancement of large language models (LLMs) has revolutionized code generation, enabling models to achieve near-human performance on programming tasks (Chen et al., 2021a; Li et al., 2022; OpenAI, 2023). These models have demonstrated remarkable abilities to generate syntactically correct and functionally viable code snippets, significantly aiding software development processes. Building upon these successes, recent research has explored the use of reinforcement learning (RL) from unit test feedback to further enhance the code generation capabilities of LLMs (Le et al., 2022; Shojaee et al., 2023; Liu et al., 2023; Dou et al., 2024). By incorporating unit tests as a reward mechanism, these methods aim to guide LLMs toward generating code that not only compiles but also passes specified test cases, thereby improving overall code reliability and quality.

However, a significant challenge arises from the nature of the reward signals derived from unit tests. These signals are inherently sparse, as they are only received at the end of an episode after the entire code snippet has been generated and evaluated. This delay in feedback impedes learning efficiency and limits the model's ability to make incremental improvements during code generation. When an LLM fails to generate code that passes any unit tests, it receives no meaningful learning signal, making it difficult to learn to solve more complex coding problems. In contrast, human programmers typically do not rewrite code from scratch when their programs fail unit tests. Instead, they analyze the code to pinpoint and fix errors, leveraging their understanding of programming logic and structure to iteratively improve upon the current version. This process of step-by-step refinement, which involves receiving and acting upon fine-grained feedback, is missing in the current RL training loop for code generation from unit test feedback.

To address this limitation, we propose integrating a Process Reward Model (PRM) (Lightman et al., 2023; Wang et al., 2024a) into the RL training framework for code generation. A PRM provides dense signals by offering line-level feedback that indicates the correctness of each generated line of code. This fine-grained feedback mechanism mimics the human approach to code refinement and has the potential to enhance learn-

ing efficiency by providing immediate guidance during code generation. While the concept of using PRMs is intuitive, training an effective PRM and integrating it into RL training is non-trivial. Challenges include accurately modeling the correctness of partial code snippets and ensuring stable and effective training dynamics when combining PRM-generated signals with traditional RL methods. Although previous research has attempted to incorporate PRMs into LLM RL training (Wang et al., 2024a), these efforts have been limited to the mathematical domain and have not fully explored the complexities involved.

In this work, we conduct a comprehensive analysis of how PRMs can be integrated into RL training for code generation. We explore various strategies for training a robust code PRM and investigate different methods of utilizing PRMs to improve code generation performance. Based on our experiments, we provide a practical recipe for successfully using PRMs and integrating them into RL training in the context of code generation problems. Notably, one of our key findings is that using PRMs concurrently as both dense rewards and value function initialization in RL training leads to a significant performance improvement. Our contributions can be summarized as follows:

- We propose an effective approach that automatically generates process-level supervision data by identifying the first error line in generated code using binary search. We then train a PRM on this data to generate dense signals during RL training. To the best of our knowledge, we are the first to demonstrate that PRMs can benefit RL from unit test feedback in code generation.

- We conduct systematic experiments to determine how to properly and effectively integrate PRMs into RL. Our analysis explores various strategies for training a high-quality code PRM and utilizing PRMs to improve code generation. We summarize our findings into a practical recipe for successfully using PRMs in the context of code generation.

- By following the recipe, we significantly improve our in-house proprietary LLM's pass rate from 28.2% to 29.8% on the LiveCodeBench dataset and from 31.8% to 35.8% on our in-house benchmark. Besides, we find that integrating PRMs into RL training benefits code generation in long-horizon scenarios.

## 2 PROBLEM FORMALIZATION

In code generation tasks, we define a code generation problem as a sequence of tokens $\mathbf{x} = (x_1, x_2, \ldots, x_m)$, where each $x_i$ denotes the $i$-th element or token of the input prompt, which may include problem descriptions. The primary objective for the model in this context is to process the given input $\mathbf{x}$ and generate a coherent and syntactically correct sequence of code tokens. This sequence is denoted as $\mathbf{y} = (\mathbf{y}^{(1)}, \mathbf{y}^{(2)}, \ldots, \mathbf{y}^{(T)})$, where $T$ represents the total number of code generation steps. Each individual code generation step, $\mathbf{y}^{(t)}$, $t = 1, 2, \ldots, T$, is composed of a series of tokens $y_1^{(t)}, y_2^{(t)}, \ldots, y_{n_t}^{(t)}$, where $y_i^{(t)}$ corresponds to the $i$-th token within the $t$-th step, and $n_t$ denotes the number of tokens in this step.

Typically, a pre-trained language model (LM), denoted as $p_\theta$, is employed to model the conditional probability distribution of the code generation steps $\mathbf{y}$, given the code generation problem $\mathbf{x}$, which is mathematically represented as $p_\theta(\mathbf{y} \mid \mathbf{x})$, parameterized by $\theta$. The model is optimized through training on a dataset $\mathcal{D}_{\mathbf{xy}}$ containing pairs of prompts and their corresponding code solutions. This training process, often referred to as Supervised Fine-Tuning (SFT), involves maximizing the log-likelihood of the dataset.

### 2.1 BASELINE METHOD: REINFORCEMENT LEARNING FROM UNIT TEST FEEDBACK

Code generation tasks can be formulated within a Reinforcement Learning (RL) framework, where code generation is treated as a sequence of decision-making steps. Once the model has undergone SFT, the RL phase is employed to refine the model's ability to generate functionally correct code using feedback from unit tests (Liu et al., 2023). Unit test feedback is derived by executing the generated program on predefined test cases. The feedback serves as a signal for learning and can be transformed into a reward. A simple

reward function based on the outcome of the unit tests could be defined as follows:

$$R_{\text{UT}}(\mathbf{x}, \mathbf{y}) = \begin{cases} 1, & \text{if the program } \mathbf{y} \text{ passes all unit test cases} \\ 0, & \text{otherwise} \end{cases}$$

This binary reward formulation encourages the model to generate programs that can successfully pass all unit test cases. Given a collection of unlabeled code generation prompts $\mathcal{D}_{\mathbf{x}}$, the model $p_\theta$ is optimized to maximize the expected reward over all possible code generation trajectories.

## 3 PROCESS SUPERVISION-GUIDED POLICY OPTIMIZATION

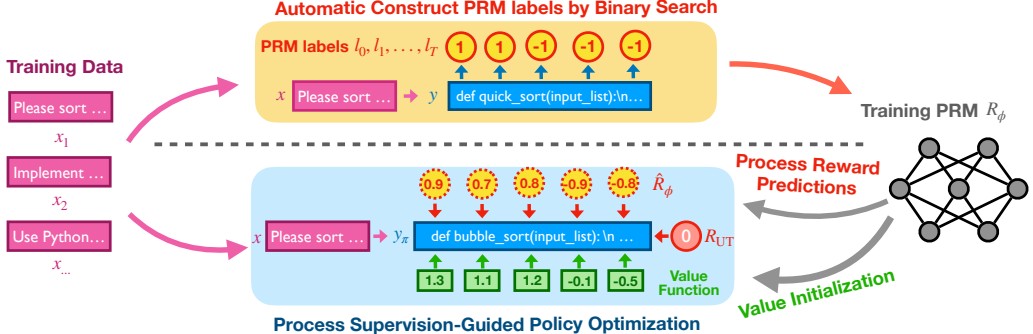

Figure 1: Overview of our method. The approach consists of two main components: (1) A binary search-based method to automate PRM training data labeling, which is used to train a code PRM; and (2) Integration of the PRM into RL training as both dense rewards and value function initialization.

While the Reinforcement Learning from Unit Test Feedback (RLTF) offers a framework for improving code generation models, it suffers from significant limitations due to the sparsity of its reward signal. The binary nature of unit test feedback—indicating only whether the entire program passes or fails—provides no guidance on which specific parts of the code contributed to the outcome. This lack of intermediate feedback makes it challenging for the model to identify and correct errors during training, leading to slow convergence and suboptimal performance. In contrast, human programmers iteratively develop and refine their code. When a program fails to pass unit tests, they do not typically rewrite it from scratch. Instead, they analyze the code to pinpoint and fix errors, leveraging their understanding of programming logic and structure. This process of step-by-step refinement is crucial for efficient problem-solving.

Motivated by this observation, we propose **Process Supervision-Guided Policy Optimization**, a method that integrates fine-grained feedback into the RL framework. Figure 1 illustrates the overview of our approach. By providing intermediate rewards that assess the correctness of partial code sequences, our approach guides the model more effectively toward generating functionally correct programs. This is achieved through a Process Reward Model (PRM) (Lightman et al., 2023) that evaluates each code generation step, offering dense reward signals that addresses the limitations of sparse end-of-trajectory rewards.

### 3.1 PROCESS SUPERVISION VIA PROCESS REWARD MODELS

Our method introduces a PRM to assess the correctness of each **line** of the code during the generation process. The PRM serves as an oracle that provides intermediate rewards based on the potential of the current code prefix to be extended into a correct program. By offering intermediate feedback, the PRM helps the model identify and reinforce beneficial code generation patterns while discouraging those that introduce errors. This fine-grained feedback mirrors the human approach to coding, where programmers continuously evaluate and adjust their code.

### 3.1.1 DATA COLLECTION

To effectively train the PRM, we require a dataset that provides fine-grained annotations indicating the correctness of partial code sequences. However, manually annotating the correctness of each line of code generated by the model is costly and not scalable. Instead, we employ an automated approach inspired by techniques used in recent works (Wang et al., 2024a;b; Luo et al., 2024). Our method leverages the model's own capabilities to generate completions for partial code prefixes and uses automated testing to assess their correctness. The key idea is to determine whether a partial code prefix can be extended—by any means—into a complete program that passes all unit tests. If so, we consider the prefix as potentially correct; otherwise, it is labeled as incorrect.

Given a prompt $\mathbf{x}$, we generate a complete code response $\mathbf{y} = (\mathbf{y}^{(1)}, \mathbf{y}^{(2)}, \ldots, \mathbf{y}^{(T)})$ using the current policy $p_\theta$. Our goal is to determine the correctness of each partial code prefix $\mathbf{y}^{\leq t}$ for $t = 1, 2, \ldots, T$. To achieve this, we employ a *best-of-$K$* sampling strategy to approximate an oracle capable of completing partial code prefixes. For each partial code prefix $\mathbf{y}^{\leq t}$, we generate $K$ potential completions $\{\mathbf{c}_k\}_{k=1}^{K}$ using the current policy. We then form full programs $\mathcal{P}_k = (\mathbf{y}^{\leq t}, \mathbf{c}_k)$ and execute them against the unit tests $\mathcal{U}$. If any of these programs pass all unit tests, we label the partial code prefix as **correct**; otherwise, it is labeled as **incorrect**. To efficiently identify the transition point where errors occur, we employ a binary search over the code generation steps (Luo et al., 2024), which is formalized in Algorithm 1. For example, consider a code response divided into five steps ($T = 5$), as shown in Figure 2. The partial prefix up to $\mathbf{y}^{(3)}$ can be completed to pass all unit tests, so it is labeled as correct. The prefix up to $\mathbf{y}^{(4)}$ cannot, meaning steps beyond $\mathbf{y}^{(3)}$ are labeled as incorrect. For each partial code prefix $\mathbf{y}^{\leq m}$, the label $l_m$ is assigned based on the outcome of the completion attempts:

$$l_m = \begin{cases} +1, & \text{if any } \mathcal{P}_k \text{ passes all unit tests} \\ -1, & \text{otherwise} \end{cases} \tag{1}$$

which indicate whether the prefix is potentially correct (can be completed to a correct program) or incorrect (contains unrecoverable errors).

### 3.1.2 TRAINING

Using the collected data $\{(\mathbf{x}, \mathbf{y}^{\leq m}, l_m)\}$, we train the PRM $R_\phi$ to predict the correctness of partial code prefixes. The PRM learns to assign higher rewards to prefixes labeled as correct and lower rewards to those labeled as incorrect. The training objective, i.e., Mean Squared Error (MSE), minimizes the discrepancy

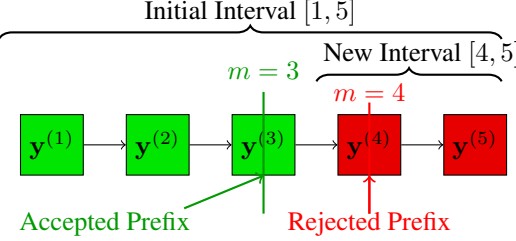

Figure 2: Binary search over code steps at line level to label prefixes. The first midpoint at $m = 3$ is accepted, so the search interval moves to $[4, 5]$. The next midpoint at $m = 4$ is rejected, indicating unrecoverable errors occur after step 3.

---

**Algorithm 1:** Binary Search for Labeling Partial Code Prefixes

---

**Input:** Prompt $\mathbf{x}$, response
    $\mathbf{y} = (\mathbf{y}^{(1)}, \ldots, \mathbf{y}^{(T)})$, policy $p_\theta$, unit
    tests $\mathcal{U}$, number of completions $K$
**Output:** Labels $l_t$ for each prefix $\mathbf{y}^{\leq t}$
Initialize lower bound $L \leftarrow 1$, upper bound
  $R \leftarrow T$, failure point $F \leftarrow T + 1$;
**while** $L \leq R$ **do**

    Compute midpoint $m \leftarrow \left\lfloor \dfrac{L + R}{2} \right\rfloor$;

    Set success flag $S \leftarrow$ False;
    **for** $k = 1$ *to* $K$ **do**
        Generate completion $\mathbf{c}_k \sim p_\theta(\cdot \mid \mathbf{y}^{\leq m})$;
        Form full program $\mathcal{P}_k \leftarrow (\mathbf{y}^{\leq m}, \mathbf{c}_k)$;
        Execute $\mathcal{P}_k$ with unit tests $\mathcal{U}$;
        **if** $\mathcal{P}_k$ *passes all unit tests* **then**
            Set $S \leftarrow$ True; **break**;

    **if** $S = True$ **then** $L \leftarrow m + 1$ ;
    **else** $F \leftarrow m, R \leftarrow m - 1$ ;
**for** $t = 1$ *to* $T$ **do**
    **if** $t < F$ **then** $l_t \leftarrow +1$;
    **else** $l_t \leftarrow -1$;

---

between the PRM's predictions and the assigned labels:

$$\min_{\phi} \sum_{(\mathbf{x}, \mathbf{y}^{\leq m})} \left( R_{\phi}(\mathbf{x}, \mathbf{y}^{\leq m}) - l_m \right)^2 \tag{2}$$

This regression formulation allows the PRM to estimate the likelihood that a given prefix can be successfully completed. Notably, aside from employing Mean Squared Error (MSE) loss, we also experimented with Cross-Entropy Loss and empirically found that MSE loss yielded better performance in our case.

## 3.2 INTEGRATING PRM INTO RL TRAINING

Given a learned PRM, we aim to identify best practices for enhancing code generation during RL training. While prior work has used PRMs to verify intermediate steps in mathematical tasks (Lightman et al., 2023; Wang et al., 2024a; Jiao et al., 2024; Wang et al., 2024b; Luo et al., 2024), their potential for guiding code generation remains largely unexplored. In mathematical domains, LLMs may generate correct answers with faulty reasoning (Lightman et al., 2023), making intermediate verification essential. However, in code generation, problems are typically accompanied by multiple unit tests, making it improbable for incorrect code to pass all tests. As a result, the emphasis on intermediate verification is less applicable. Instead, we propose leveraging PRMs as auxiliary sources of dense signals to facilitate better exploration during RL training. While preliminary attempts have been made to incorporate PRMs into RL training (Wang et al., 2024a), these efforts are limited and warrant a more thorough investigation. We explore the following methods to integrate PRMs effectively:

**PRM as Dense Rewards.** Similar to Wang et al. (2024a), we use PRMs to provide step-level reward signals that guide more efficient policy exploration during RL training. By rating the correctness of each line in the code response, the PRM supplies "dense" rewards that encourage the policy to explore more promising code paths, leading to improved performance.

**PRM as Value Initialization.** The PRM's method of annotating code, by fixing a prefix $\mathbf{y}^{\leq t}$ and rolling out the policy to sample correct responses, can be viewed as a "hard" value estimation of $\mathbf{y}^{\leq t}$. We hypothesize that the PRM's capability to provide line-level feedback could serve as a useful inductive bias for initializing the value function in RL algorithms, which can ease the credit assignment problem by offering a more informed starting point.

**PRM as Both Dense Rewards and Value Initialization.** To fully capitalize on the advantages of PRMs, we combine both approaches. By using PRMs for dense rewards and as an initialization for the value model, we aim to enhance RL training through improved exploration and more effective credit assignment.

## 4 EXPERIMENTAL RESULTS

### 4.1 EXPERIMENTAL SETUP

**Datasets and Evaluation.** We utilize in-house datasets to train our model for code generation. Specifically, the training set, $\mathcal{D}_{\text{train}}$, is a comprehensive Reinforcement Learning with Human Feedback (RLHF) dataset that includes, as a subset, approximately $30,000$ diverse coding problems. Each of these problems is paired with unit tests designed to validate the functional correctness of the generated code. For evaluation, we employ two benchmarks: the publicly available LiveCodeBench (Jain et al., 2024) and our proprietary code generation benchmark (InHouseBench). LiveCodeBench is a comprehensive benchmark designed to evaluate the code generation capabilities of LLMs. Among its various releases, we used LiveCodeBench v3, which consists of 612 coding tasks collected between May 2023 and July 2024. InHouseBench comprises 245 challenging coding problems in Chinese, spanning both Python and C++. All test problems are novel and unseen in public datasets, ensuring there is no risk of data contamination. The benchmark contains two problem categories: (1) Contest, which includes 169 coding competition-style problems, and (2) NL2Alg, which comprises 76 problems focused on translating natural language algorithm descriptions in Chinese into

executable code. For evaluation, we let each model generate 10 candidate responses for each problem, using a temperature of 0.2, nucleus sampling with top-$p$=0.95, and top-k sampling with $k$=128, following common practice. We adopt Pass@1 as the evaluation metric, in line with previous work (Kulal et al., 2019; Chen et al., 2021a; Jain et al., 2024).

**Base Models and RL Baseline.** The base model used in our experiments is a small in-house model, referred to as `InHouse-Lite` throughout the remainder of this paper. Initially, `InHouse-Lite` was fine-tuned on our Supervised Fine-Tuning (SFT) dataset, resulting in `InHouse-Lite-SFT`, which then served as the initialization for the subsequent RLHF training phase. We fine-tune `InHouse-Lite-SFT` ($\pi_{\mathrm{ref}}$) on the RLHF dataset $\mathcal{D}_{\mathrm{train}}$ using Proximal Policy Optimization(PPO) (Schulman et al., 2017) to obtain `InHouse-Lite-RL` ($\pi_\theta$). In our setup, two types of Outcome Reward Models (ORMs) are employed as the objective functions for RL training. For non-coding prompts, we use a general reward model, $R_{\mathtt{general}}(\mathbf{x}, \mathbf{y})$, derived from preference learning on a human-annotated dataset (Ouyang et al., 2022). For coding prompts, the ORM is defined as a binary indicator of whether the response passes all unit tests, $R_{\mathrm{UT}}$ ($\mathbf{x}, \mathbf{y}$). Following Ouyang et al. (2022), RLHF optimization objective is defined as:

$$\max_\theta \sum_{\mathbf{x} \in \mathcal{D}_{train}} \mathbb{E}_{\mathbf{y} \sim \pi_\theta(\mathbf{y}|\mathbf{x})} \left[ R(\mathbf{x}, \mathbf{y}) - \beta \mathrm{KL}(\pi_\theta \parallel \pi_{\mathrm{ref}}) \right],$$

with $R(\mathbf{x}, \mathbf{y}) = R_{\mathtt{general}}(\mathbf{x}, \mathbf{y})$ for non-coding prompts and $R(\mathbf{x}, \mathbf{y}) = R_{\mathrm{UT}}(\mathbf{x}, \mathbf{y})$ for coding prompts.

**PRM Training.** To ensure that the PRM training data effectively covers the state space the language model may encounter during the next RL training phase, we sample policy models from various stages of the RL baseline training. Specifically, we select 4 checkpoints evenly spaced throughout the RL baseline model's training process. For each checkpoint, we sample $n$ responses for each coding prompt in the training dataset $\mathcal{D}_{\mathrm{train}}$. For each sampled response, we apply the binary search labeling procedure described in Algorithm 1, using $K = 20$ completions for each partial code prefix. The data collected from all checkpoints is then aggregated into a PRM training set, denoted as $\mathcal{D}_{\mathrm{PRM}}$. We initialize the PRM with the value model from the RL baseline and fine-tune it on the aggregated dataset, $\mathcal{D}_{\mathrm{PRM}}$, using the objective function defined in Eq. (2).

**Integrating PRM into RL.** As described in Section 3.2, we explore two methods for integrating the Process Reward Model (PRM) into RL training: (1) using PRM as a source of dense reward signals (**DenseReward**) and (2) initializing the value function in PPO with PRM (**ValueInit**). In the **DenseReward** approach, PRM assigns additional reward signals at each end-of-line token ($\backslash$n) in the code response for coding prompts. Thus, the RL optimization objective for coding prompts is modified to the weighted sum of $R_{\mathrm{UT}}$ and $R_{\mathrm{PRM}}$, as defined below:

$$\max_\theta \sum_{\mathbf{x} \in \mathcal{D}_{train}} \mathbb{E}_{\mathbf{y} \sim \pi_\theta(\mathbf{y}|\mathbf{x})} \left[ R_{\mathrm{UT}}(\mathbf{x}, \mathbf{y}) + \lambda R_{\mathrm{PRM}}(\mathbf{x}, \mathbf{y}) - \beta \mathrm{KL}(\pi_\theta \parallel \pi_{\mathrm{ref}}) \right], \tag{3}$$

where $\lambda$ controls the relative importance of PRM in shaping the reward. Specifically, we set $\lambda = 0.25$ when the code response does not pass all unit tests, i.e., $R_{\mathrm{UT}}(\mathbf{x}, \mathbf{y}) = 0$, and $\lambda = 0.025$ when the response passes all unit tests, i.e., $R_{\mathrm{UT}}(\mathbf{x}, \mathbf{y}) = 1$. The intuition behind this reward shaping is to leverage PRM to provide informative signals when the RL policy fails to generate a valid solution, while minimizing the risk of PRM over-optimization (Rafailov et al., 2024; Skalse et al., 2022) once a correct solution is found. Our empirical results indicate that this reward shaping strategy performs effectively in our experimental setting. In the **ValueInit** setting, PRM is simply used to initialize of the value function in PPO. Notably, these two approaches-**DenseReward** and **ValueInit**—are complementary and can be applied concurrently.

## 4.2 KEY CONSIDERATIONS FOR INTEGRATING PRM INTO RL TRAINING

While integrating PRM into RL training might seem straightforward, we found that achieving effective results requires careful attention to several critical factors. In this section, we highlight key implementation details essential for the successful application of PRM in RL training.

### 4.2.1 PRM TRAINING: MORE DATA OR BETTER DATA?

Recent research on LLMs highlights that data quality often outweighs quantity (Gunasekar et al., 2023; Li et al., 2023b). We found the same holds true for PRM training data selection. Although automated data annotation allows for the generation of large volumes of PRM training data through model sampling, our experiments showed that increasing data volume can sometimes degrade PRM performance when integrated into RL. In contrast, a smaller, carefully curated subset of the full dataset led to better supervision and improved outcomes. For example, when all sampled responses to a given prompt either consistently pass (or fail) unit tests, PRM gains little useful information. In such cases, the model can only learn to predict the correct (or incorrect) label when it encounters the same prompt again, limiting its ability to generalize. We explored various strategies for selecting and filtering data, as detailed in Section 4.3.

### 4.2.2 RL TRAINING: ALLEVIATING PRM HACKING

Reward model hacking (Skalse et al., 2022) is a well-known issue in RLHF training, where the policy learns to exploit flaws in the reward model to achieve high rewards without genuinely improving the quality of response. Similarly, we observed that PRM is also susceptible to such exploitation. Here we discuss two key practical strategies to mitigate the risk of PRM hacking and ensure the reward signals remain aligned with the intended task objectives.

**PRM reward length normalization.** As described in Section 4.1, when used to provide dense rewards, PRM assigns line-level reward signals at the end-of-line tokens in the LLM-generated response. However, if we directly use the predictions of the learned PRM, denoted as $R_\phi$, as the reward signal $R_{\text{PRM}}$ in 3, we observed that this can be exploited. Specifically, the policy may generate numerous lines for which PRM predicts positive rewards, thus inflating the overall reward. This occurs because writing more lines allows the model to accumulate excessive intermediate rewards, effectively hacking the optimization objective. To mitigate this issue, we apply length normalization to the PRM predictions. Given a prompt $\mathbf{x}$ and a response $\mathbf{y}$ with $T$ lines, $\mathbf{y} = (\mathbf{y}^{(1)}, \mathbf{y}^{(2)}, \ldots, \mathbf{y}^{(T)})$, we define the PRM dense reward signal at the $m$-th line as:

$$R_{\text{PRM}}(\mathbf{y}^{(m)}) = \frac{1}{T} \cdot R_\phi(\mathbf{x}, \mathbf{y}^{\leq m}).$$

This normalization ensures that the policy does not gain higher cumulative rewards by generating trivial or unnecessarily long responses, as the accumulated reward is bounded within the range of $[-1, 1]$ regardless of the response length.

**Assigning additional neutral label into PRM training.** While length normalization helps reduce PRM exploitation, it is insufficient to fully prevent PRM hacking. Even with normalization, we empirically observed that the model can still exploit PRM by generating excessive comment lines within the code. The underlying issue is that writing a correct comment is often far easier than producing correct code. As a result, the model can artificially inflate the PRM reward by including unnecessary comment lines. To address this issue, we introduce an additional neutral label in the PRM annotation, as defined in Equation 1:

$$l_m = \begin{cases} +1, & \text{if any } \mathcal{P}_k \text{ passes all unit tests} \\ 0, & \text{if the line is a comment} \\ -1, & \text{otherwise} \end{cases}$$

By assigning a neutral label (0) to comment lines, we remove the reward bias that encourages the model to generate unnecessary comments. This adjustment ensures that only meaningful contributions to the code, rather than extraneous comments, are rewarded by the PRM.

### 4.3 MAIN RESULTS AND ANALYSIS

| Model | Setting | | Dataset | | | | | | |
|---|---|---|---|---|---|---|---|---|---|
| | Dense Reward | Value Init. | LiveCodeBench | | | | InHouseBench | | |
| | | | Easy | Medium | Hard | Overall | Contest | NL2Alg | Overall |
| GPT-4o-mini | - | - | 81.9 | 27.2 | 3.6 | 40.7 | 43.8 | 68.4 | 51.4 |
| Qwen2-72B | - | - | 65.0 | 21.3 | 2.8 | 32.2 | 14.8 | 51.3 | 26.1 |
| Gemini-Flash-1.5 | - | - | 67.7 | 13.1 | 1.9 | 29.6 | - | - | - |
| DeepseekCoder-33B | - | - | 60.8 | 14.8 | 1.2 | 27.7 | 10.3 | 50.3 | 22.7 |
| Ours-SFT | - | - | 55.3 | 9.3 | 0.3 | 23.5 | 10.4 | 41.4 | 20.0 |
| Ours-RL | × | × | **70.0** | 7.2 | 1.7 | 28.2 | 24.4 | 48.7 | 31.8 |
| | × | ✓ | 67.9 | 8.9 | 1.9 | 28.2 | 25.0 | 45.4 | 31.4 |
| | ✓ | × | 68.5 | 9.9 | **2.5** | 28.9 | 25.2 | 48.1 | 32.3 |
| | ✓ | ✓ | 69.3 | **12.0** | 1.6 | **29.8** | 27.9 | 53.5 | 35.8 |

Table 1: Comparison of model performance (Pass@1) across LiveCodeBench and InHouseBench with PRM as DenseReward and ValueInit settings. The performance of our models (`InHouse-Lite` series) on both LiveCodeBench and InHouseBench are averaged over 10 independent runs. We also report the performance of other public models for comparative purpose. The performance of Gemini-Flash-1.5 on InHouseBench is omitted due to legal considerations.

**Comparing Different Strategies of Using PRM in RL Training.** We explore three strategies for integrating the PRM into RL training, as described in Section 4.1: DenseReward, ValueInit, and a combined approach of DenseReward & ValueInit. The performance of RL models trained using these strategies on LiveCodeBench and InHouseBench, in comparison to SFT and RL baselines, is summarized in Table 1. For further comparison, we also include results from several publicly available models OpenAI (2023); Bai et al. (2023); Reid et al. (2024); Guo et al. (2024). Our experimental results reveal that using PRM solely as dense rewards significantly outperforms the RL baseline (Our-RL without DenseReward and ValueInit in Table 1), consistent with findings from Wang et al. (2024a). This suggests that the granular feedback provided by PRM helps the policy explore more promising solutions by offering continuous corrections at intermediate steps. Moreover, we observe that combining PRM as both dense rewards and value function initialization results in substantial performance gains, with a relative increase of 5.7% on LiveCodeBench and 12.6% on InHouseBench compared to the RL baseline.

Interestingly, using PRM solely for value function initialization does not provide notable benefits. We hypothesize that while value function initialization can enhance credit assignment and stabilize the learning process, it does not address the underlying issue of sparse reward signals. Without dense feedback, the policy may fail to explore the solution space effectively, resulting in limited

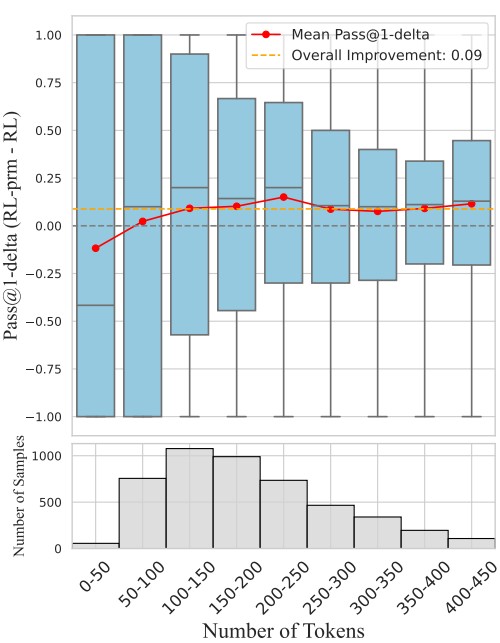

Figure 3: Pass@1 difference between policies trained with and without PRM across varying response lengths. Policies trained with PRM exhibit consistent improvements over those without PRM for longer-horizon responses (greater than 100 tokens). This demonstrates PRM's effectiveness in providing intermediate feedback, thereby enabling RL to do more explorations.

improvement. In contrast, the combination of DenseReward and ValueInit offers a synergistic effect: dense rewards enhance exploration by providing rich intermediate feedback, while value function initialization improves stability and credit assignment. Together, these mechanisms enable the policy to converge more efficiently toward optimal solutions, which explains the significant performance improvements we observed.

**PRM enhances code generation in long-horizon scenarios.** To better understand when PRM benefits code generation most, we analyze its effect based on the length of the generated responses. Intuitively, the dense nature of the reward signals provided by PRM is particularly advantageous for long-horizon tasks, where intermediate feedback can guide policy exploration more effectively. To validate this, we quantitatively compared the pass rate (Pass@1) of models trained with and without PRM across different response lengths. The results are visualized in Figure 3. Overall, the model trained with PRM demonstrates a 9% improvement in Pass@1 compared to the baseline model trained without PRM. Notably, PRM consistently improves Pass@1 for responses longer than 100 tokens. However, for responses shorter than 100 tokens, PRM yields comparable or slightly inferior results. We hypothesize that in short-horizon scenarios, PRM may function similarly to a biased ORM, which limits its ability to provide meaningful improvements. Shorter responses inherently benefit less from the dense feedback signals PRM offers, as these tasks may already be well-explored by the policy without needing extensive guidance. In contrast, for more complex, long-horizon tasks, PRM offers valuable intermediate signals that effectively guide the policy to explore better solutions. These signals provide a more nuanced understanding of the correctness of individual code lines, helping the policy navigate the larger solution space with the same amount of optimization compute.

**The Importance of PRM Training Data Selection** PRM training data can be categorized at two levels: At the *response level*, responses are classified as Correct (passes unit tests immediately), Revised (initially fails but can find a correct prefix), and Wrong (cannot find any correct prefix by binary search within the given budget). At the *prompt level*, prompts are categorized as **Easy** (all responses are Correct), **Medium** (mixed response types), and **Hard** (all responses are Wrong). We tested the following data selection strategies: **Full** (use all collected data), **Remove Hard** (exclude Hard prompts and their responses), **Medium Only** (include only prompts with mixed response types), and **Revised Only** (use only Revised responses). We empirically found that **Revised Only**, which includes the richest process-level correction signals, performs best in our setting.

| Strategy | LCB | IHB |
|---|---|---|
| **Full** | 26.9 | 34.6 |
| **Remove Hard** | 27.8 | 33.8 |
| **Medium Only** | 26.9 | 32.5 |
| **Revised Only** | 29.8 | 35.8 |

Table 2: Comparison of different PRM data selection strategies on two datasets: **LCB** (LiveCodeBench) and **IHB** (In-HouseBench).

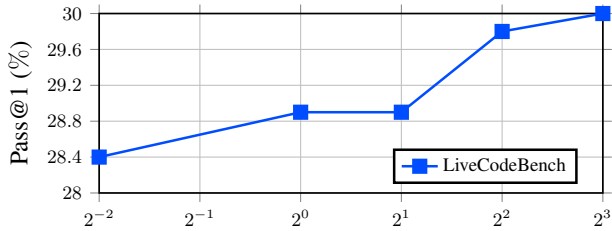

Figure 4: Pass@1 on LiveCodeBench as the average number of responses per prompt for PRM data collection increases (logarithmic scale).

**How much data needed to train a PRM that benefits RL training?** Given that automatic PRM data collection is computationally expensive, we examine how the performance of policies trained with PRM scales with the number of training samples. Figure 4 shows how the pass rate of models trained with varying amounts of PRM data changes along the average number of responses collected per prompt for PRM data collection, as described in Section 3.1.1. The key finding is that the performance of models trained with PRM improves consistently as the number of PRM training samples increases, highlighting the effectiveness and scalability of our approach.

## 5 RELATED WORKS

### 5.1 LLMs FOR CODE GENERATION

Recently, large language models (LLMs) have demonstrated impressive capabilities in code generation by pre-training on vast text datasets that include code (Lu et al., 2021; Christopoulou et al., 2022; Allal et al., 2023; Zheng et al., 2024; Li et al., 2023b). Additionally, models fine-tuned through supervised fine-tuning (SFT) have achieved competitive results in code generation tasks (Chen et al., 2021a; Li et al., 2023a; Luo et al., 2023; Rozière et al., 2024; Guo et al., 2024). Reinforcement Learning (RL) optimizes policies by interacting with an environment and receiving rewards (Williams, 1992). Recently, RL has been incorporated into LLMs to enhance code generation using unit test feedback (Shojaee et al., 2023; Liu et al., 2023; Le et al., 2022). CodeRL (Le et al., 2022) applies unit test signals as rewards with an actor-critic method, while PPOCoder (Shojaee et al., 2023) builds on this by using the PPO algorithm. RLTF (Liu et al., 2023) improves precision by locating errors, though the reward space remains sparse. Despite progress, RL's potential to significantly boost code generation in sparse reward environments remains underexplored.

### 5.2 PROCESS REWARD MODELS

Process reward models (PRMs) have garnered significant attention in recent LLM developments, particularly in the mathematical reasoning domain, where they provide verification for intermediate reasoning steps (Lightman et al., 2023; Wang et al., 2024a; Jiao et al., 2024; Wang et al., 2024b; Luo et al., 2024). While some approaches rely on costly and resource-intensive human-annotated process data (Lightman et al., 2023), recent research has focused on automating the collection of process supervision data Wang et al. (2024a); Jiao et al. (2024); Wang et al. (2024b); Luo et al. (2024). Building on these efforts, we similarly automate process supervision but differ in our primary objective. Rather than using PRMs solely as enhanced verifiers compared to Outcome Reward Models (ORMs), we focus on their integration into RL training for code generation. While Wang et al. (2024a) provides preliminary results on PRMs improving RL training in the mathematical domain, their findings are limited. Our work offers a more thorough and systematic investigation of how PRMs can be leveraged in RL for code generation tasks.

## 6 CONCLUSIONS AND LIMITATIONS

In this work, we addressed the challenge of sparse reward signals in reinforcement learning (RL) for code generation by introducing a Process Reward Model (PRM) that provides dense, line-level feedback. This approach mirrors human-like code refinement and improves learning efficiency. Our experiments demonstrate that integrating PRMs significantly enhances the pass rates of code generation models on both the Live-CodeBench dataset and proprietary benchmarks. This method has the potential to improve long-horizon code generation scenarios, advancing the state-of-the-art in LLM-based code generation.

Despite the promising results, our approach has several limitations that warrant further investigation. First, the effectiveness of the PRM relies heavily on the quality of the collected data. While we automated data collection using binary search and unit tests, this method may not capture all nuances of code correctness and may introduce noise, especially in more complex or ambiguous programming tasks. Second, the computational cost of collecting PRM training data is still substantial, even though we employed binary search to mitigate it. Third, our automated PRM data collection method requires external verification (unit tests in our case), which is not applicable to many other domains such as creative writing or open-ended generation problems. This could limit the applicability of our approach in those areas.

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

## A IN-HOUSE LLM EXPERIMENT DETAILS: PRM TRAINING DATA STATISTICS

We provide detailed statistics for the PRM datasets used in the experiments to determine the best PRM data selection strategy, as discussed in Section 4.3. Table 3 summarizes the following key metrics: the number of prompt-response pairs (**#Samples**); the total number of tokens across all responses (**#Tokens**); the average number of lines in all responses (**Avg. #Lines**); and the distribution of PRM labels ($-1/0/+1$).

In Figure 5, we present the distribution of error positions of all Revised responses (responses that initially fail but have a correct prefix identified) as determined by the Binary Search procedure (Algorithm 1). The absolute error position (i.e., the position of the first token rejected by Binary Search) is normalized as follows: for a response $\mathbf{y} = (y_1, y_2, \ldots, y_L)$ with $L$ tokens, if the Binary Search accepted the prefix $(y_1, y_2, \ldots, y_p)$ consisting of $p$ tokens, the Relative Error Position is calculated as $\frac{p}{L}$.

| Strategy | #Samples | #Tokens | Avg. #Lines | PRM Labels | | |
|---|---|---|---|---|---|---|
| | | | | $-1$ | $0$ | $+1$ |
| **Full** | 838K | 179M | 16.82 | 44.25% | 17.87% | 37.88% |
| **Remove Hard** | 630K | 119M | 15.06 | 24.03% | 19.18% | 56.79% |
| **Medium Only** | 485K | 104M | 16.57 | 27.66% | 19.41% | 52.93% |
| **Revised Only** | 352K | 76M | 16.71 | 13.20% | 19.42% | 67.38% |

Table 3: Statistics of PRM training data collected using different data selection strategies.

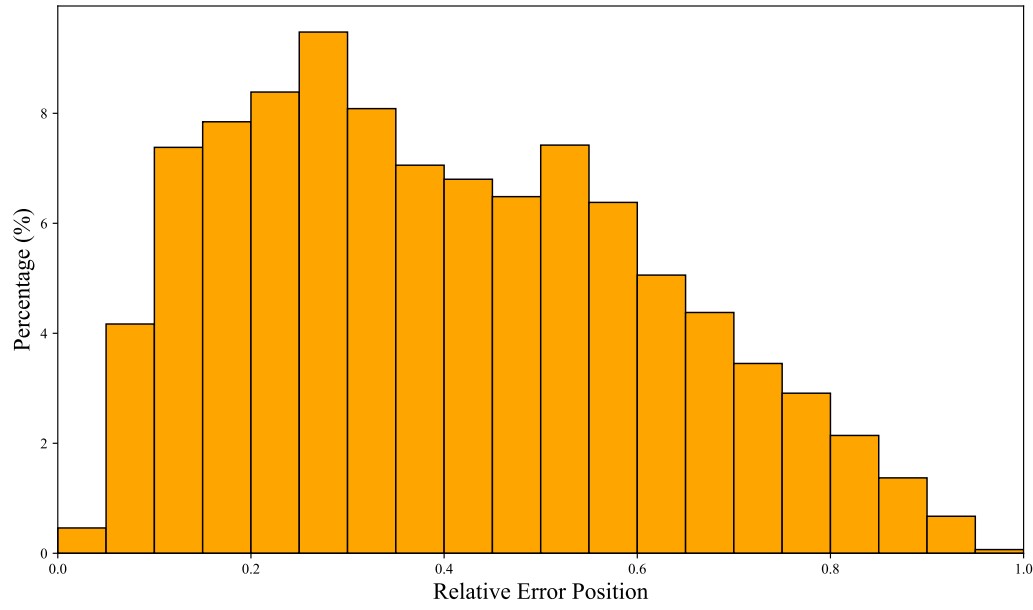

Figure 5: Distribution of Relative Error Positions Identified by Binary Search.

# B  IN-HOUSE LLM EXPERIMENT DETAILS: RL TRAINING CURVES

In Figure 6, we present the smoothed RL training curves for all four settings (with and without DenseReward, and with and without ValueInit) using a moving average to reduce noise and enhance readability. These curves correspond to all four RL settings reported in Table 1. The smoothed trends clearly show that when PRM is used as DenseReward, the model solves more problems compared to the baseline, demonstrating PRM's role in enabling more efficient exploration during RL training. Furthermore, when PRM is applied as both DenseReward and ValueInit, our method achieves the best performance.

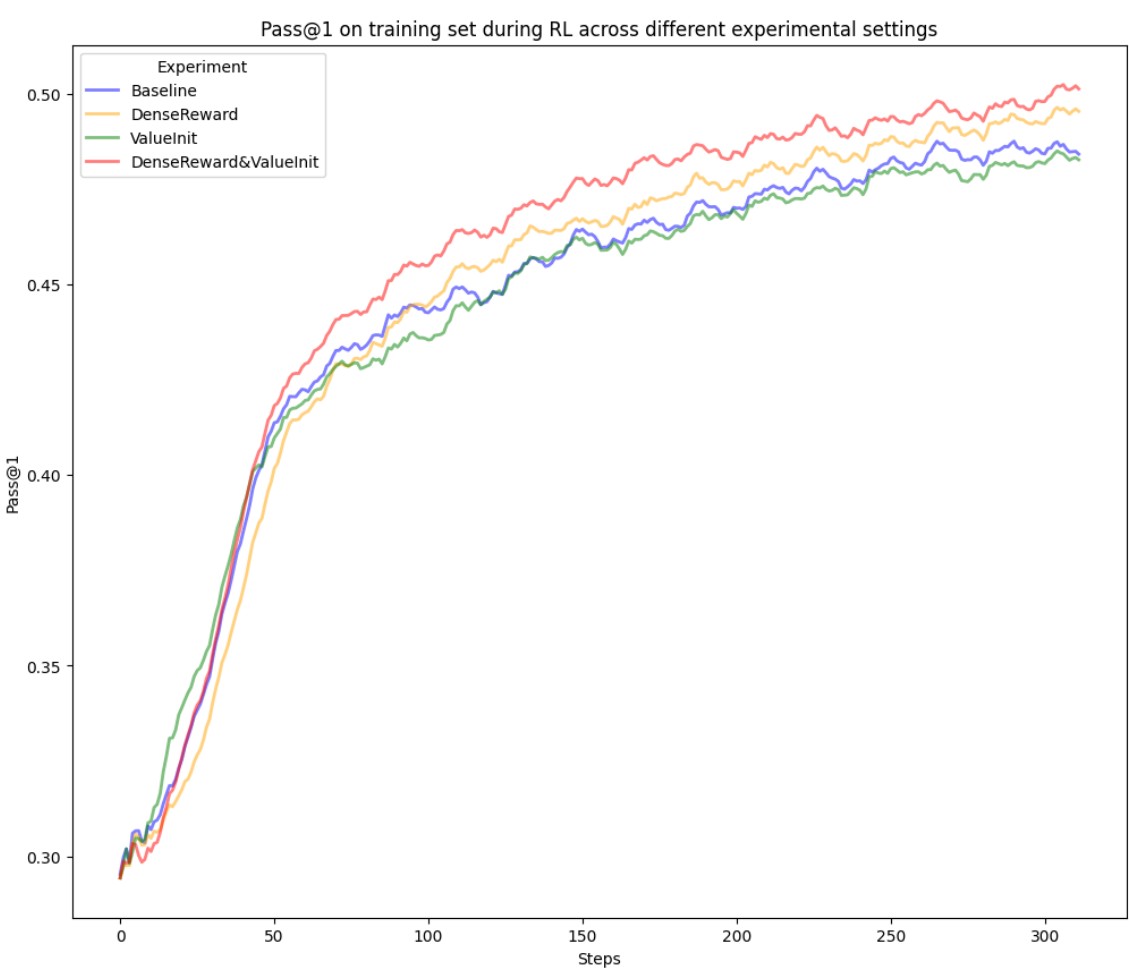

Figure 6: RL training curve of our method (DenseReward&ValueInit) compared to other settings. Using PRM as both DenseReward and ValueInit yields the best result.

In addition, we evaluated the Best-of-K performance for all four settings on the training set. Specifically, we used the checkpoint at 300 steps for each setting and evaluated their Best-of-K performance using a decoding configuration with a temperature of 1.0, nucleus sampling with top-$p$=0.95, and top-k sampling with $k$=128. For each K, we recorded the percentage of problems that the model solved within K generated responses, which we refer to as the *Pass Rate*.

In Figure 7, we present the evaluation results for K ranging from 1 to 30. The plot shows that both DenseReward and ValueInit independently improve the Best-of-K performance compared to the baseline. When both DenseReward and ValueInit are enabled, the model achieves the highest boost, with an improvement in pass rate of nearly 4% at K=30 compared to the baseline.

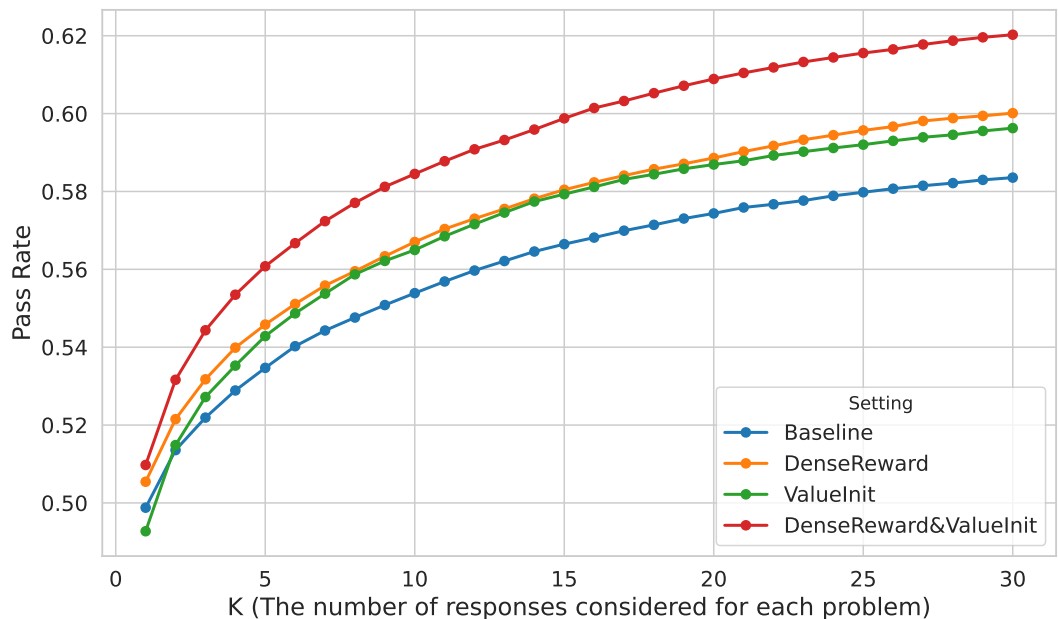

Figure 7: Best-of-K performance curves for all RL training settings, showing the percentage of problems solved within K generated responses for each configuration.

## C   IN-HOUSE LLM EXPERIMENT RESULTS ON HUMANEVAL AND MBPP

We evaluated our models on two additional coding benchmarks: HumanEval and MBPP. The descriptions of these two datasets are provided below. For evaluation, we used the same settings as described in Section 4.1. Specifically, we generated 10 candidate responses for each problem, using a temperature of 0.2, nucleus sampling with top-$p$=0.95, and top-k sampling with $k$=128, following common practice. Pass@1 was used as the evaluation metric. The results can be found in Table 4.

**HumanEval (Chen et al., 2021b)**   This dataset comprises 164 hand-crafted programming problems that are designed to evaluate the functional correctness of code generated by LLMs, rather than merely comparing textual similarity to reference solutions. The problems in HumanEval cover a range of tasks, including language comprehension, algorithms, and basic mathematics, and they are comparable to typical software interview questions. Each problem is accompanied by a function signature, a docstring, a function body, and multiple unit tests to rigorously test the generated solutions. We directly input the problem to the model without using few-shot prompting.

**MBPP (Austin et al., 2021)**   This dataset comprises 974 crowd-sourced Python programming tasks, specifically crafted to be solvable by entry-level programmers. Each problem in the MBPP dataset consists of a task description, a code solution, and three automated test cases, covering programming fundamentals and standard library functionality. The dataset is particularly valuable for evaluating a model's ability to synthesize short Python programs from natural language descriptions. For evaluation, we used the problems with IDs 11–510, totaling 500 problems, as recommended by the original dataset authors. We directly input the problem to the model without using few-shot prompting.

| Model | Setting | | Dataset | |
| --- | --- | --- | --- | --- |
| | **Dense Reward** | **Value Init.** | **HumanEval** | **MBPP** |
| Our-SFT | - | - | 59.3 | 59.9 |
| Our-RL | ✗ | ✗ | 65.1 | 61.9 |
| | ✗ | ✓ | 69.8 | 63.3 |
| | ✓ | ✗ | 70.0 | 62.1 |
| | ✓ | ✓ | **70.9** | **63.8** |

Table 4: Comparison of model performance (Pass@1) across HumanEval and MBPP with PRM as DenseReward and ValueInit settings when using our in-house model.

# D    ADDITIONAL OPEN-SOURCE LLM EXPERIMENTS

**Base Model.**    We adopt **Qwen2.5-7B** as our base model (QwenTeam, 2024), a recently released causal language model available at `https://huggingface.co/Qwen/Qwen2.5-7B`. Qwen2.5 belongs to the Qwen series of large language models (Yang et al., 2024), known for their advanced capabilities across a wide range of domains. The Qwen2.5-7B model has 7.61 billion total parameters (6.53 billion excluding embeddings) and utilizes the Transformer architecture as its core. It incorporates state-of-the-art enhancements, including Rotary Positional Embedding (RoPE), SwiGLU activation, RMSNorm, and Attention QKV bias. The model consists of 28 layers and employs 28 attention heads for queries (Q) and 4 for keys and values (KV), making it highly efficient for tasks requiring robust attention mechanisms.

**SFT Settings.**    We fine-tuned the Qwen2.5-7B model on the same supervised fine-tuning (SFT) dataset described in Section 4.1. The model was trained for two epochs, starting with a learning rate of $1 \times 10^{-7}$, which linearly increased to $2 \times 10^{-5}$ during the first 2% of the total training steps. After reaching the peak learning rate, a cosine learning rate decay schedule was applied, gradually reducing the learning rate to $2 \times 10^{-6}$ for the remainder of the training. Additionally, a constant weight decay of 0.01 was used throughout the SFT training process to regularize the model and improve generalization. The model fine-tuned through this process is referred to as **Qwen2.5-7B-SFT**.

**RL Baseline.**    We adopted the same RL baseline training method and used the same RLHF dataset described in Section 4.1 to further train the Qwen2.5-7B-SFT model. For PPO training, we configured the following hyperparameters: a batch size of 4096, a linear warmup over the first 5 steps, followed by a constant learning rate of $2 \times 10^{-6}$ for both the actor and critic, and a KL penalty of 0.01. The training utilized the AdamW optimizer and spanned approximately 300 steps, during which we empirically observed performance convergence.

**PRM Training.**    Following the approach outlined in Section 4.1, we selected four checkpoints at 50, 100, 150, and 200 steps during the training process of the RL baseline model. For each checkpoint, we sampled $n = 5$ responses for every coding prompt in the training dataset $\mathcal{D}_{\text{train}}$. Each sampled response was labeled using the binary search procedure described in Algorithm 1, with $K = 20$ completions generated for each partial code prefix. The data collected from all checkpoints was then aggregated to form a PRM training set, employing the **Revised Only** strategy described in Section 4.3. This resulted in 165K samples and 28M tokens. On average, each response contained 16.07 lines. The PRM label distribution was 25.88% for $-1$, 15.90% for $0$, and 58.22% for $+1$. The PRM was initialized using the value model from the RL baseline and fine-tuned on this PRM dataset using the objective function defined in Eq. (2).

**Integrating PRM into RL.**    We used the same settings and hyperparameters as described in Section 4.1. Additionally, we observed that due to the properties of the Qwen2.5-7B tokenizer, a newline token is not always represented as a simple `"\n"` token. Instead, the tokenizer combines other non-space characters with an ending `"\n"` to form new tokens (e.g., `":\n"`, `"):\n"`, `")\n"`, `"\n\n"`, `"())\n"`, `"]\n"`, `"()\n"`, `"():\n"`, etc.). This makes it more challenging to accurately identify line separator tokens in the model's responses.

Empirically, we addressed this challenge by selecting the 50 most frequent tokens in the PRM dataset whose corresponding token strings include `"\n"`. The full list of token ids is shown below:

{198, 510, 982, 340, 271, 2398, 921, 741, 3932, 1171, 692, 1305, 4167, 2546, 1447, 10343, 1138, 19324, 341, 5563, 9957, 382, 3407, 3646, 624, 48443, 280, 456, 2533, 3989, 1248, 5613, 8389, 8997, 698, 24135, 317, 7368, 2440, 10907, 22165, 4432, 5929, 7129, 345, 11043, 532, 4660, 21686, 14288}.

During RL training, we only applied partial rewards from PRM to these tokens.

**Main Results.** Table 5 presents the performance of Qwen2.5-7B models with various configurations on LiveCodeBench and InHouseBench. Table 6 shows the performance of these models on additional datasets, HumanEval and MBPP, as introduced in Appendix C. Across all four datasets, models employing PRM consistently outperformed the RL baseline without PRM, demonstrating the effectiveness of PRM.

| Model | Setting | | Dataset | | | | | | |
|---|---|---|---|---|---|---|---|---|---|
| | Dense Reward | Value Init. | LiveCodeBench | | | | InHouseBench | | |
| | | | Easy | Medium | Hard | Overall | Contest | NL2Alg | Overall |
| Qwen2.5-7B-SFT | - | - | 50.7 | 16.5 | 0.9 | 24.9 | 12.3 | 35.4 | 19.4 |
| Qwen2.5-7B-RL | × | × | 60.9 | 13.7 | 1.4 | 27.5 | 26.0 | 45.6 | 32.0 |
| | × | ✓ | 62.8 | **17.1** | **1.7** | 29.6 | **30.6** | 46.1 | **33.6** |
| | ✓ | × | 63.1 | 14.5 | 1.1 | 28.5 | 27.3 | 47.6 | **33.6** |
| | ✓ | ✓ | **66.3** | 15.3 | **1.7** | **30.1** | 26.1 | **48.7** | 33.1 |

Table 5: Comparison of model performance (Pass@1) across LiveCodeBench and InHouseBench with PRM as DenseReward and ValueInit settings when using Qwen2.5-7B model.

| Model | Setting | | Dataset | |
|---|---|---|---|---|
| | Dense Reward | Value Init. | HumanEval | MBPP |
| Qwen2.5-7B-SFT | - | - | 67.8 | 58.1 |
| Qwen2.5-7B-RL | × | × | 73.8 | 62.4 |
| | × | ✓ | 75.4 | 63.1 |
| | ✓ | × | **76.0** | 63.4 |
| | ✓ | ✓ | 74.3 | **65.4** |

Table 6: Comparison of model performance (Pass@1) across HumanEval and MBPP with PRM as DenseReward and ValueInit settings when using Qwen2.5-7B model.

# E   A TYPICAL EXAMPLE OF THE LEARNED LINE-WISE REWARDS

In Figure 8, we present a typical example of the line-wise rewards identified by binary search and predicted by a learned PRM to give readers a clearer understanding of our method. In this example, we first sampled a problem from the training set and used our in-house model to generate a response for it. For this generated response (which is not included in the PRM training data), we show the line-wise rewards derived from two sources:

1. **Line-wise Rewards Identified by Binary Search:** We directly applied the model to perform Algorithm 1, labeling the reward for each line.

2. **Line-wise Rewards Predicted by a Learned PRM:** We used the learned PRM to predict the rewards for each line.

```
AOR Ika wants to create a strong password that consists only of lowercase letters. AOR Ika-chan, who was given an example of $ N $ of
dangerous passwords by a friend, decided to create a password that meets all of the following conditions.

1. The length is at least one character.
2. Different from any contiguous substring of any dangerous password.
3. This is the shortest character string that meets the conditions 1 and 2.
4. This is the character string that comes to the beginning when arranged in lexicographic order while satisfying the conditions 1, 2, and 3.

Write a program to generate a strong password on behalf of AOR Ika-chan.

input

Input is given from standard input in the following format.

$ N $
$ S_1 $
$ \ vdots $
$ S_N $

* The first line is given the integer $ N $, which represents the number of strings.
* The string $ S_i $ is given to the $ N $ line from the second line.
* $ | S_i | $ is the length of the string, which is one or more characters.
* Satisfy $ 1 \ le N \ le 100,000 $.
* $ 1 \ le \ sum_ {1 \ le i \ le N} | S_i | \ le 400,000 $.
* The string contains only lowercase letters.

output

Print the answer in one line. Also, output a line break at the end.

Example

Input

5
password
login
admin
root
master

Output

b
```

**Line-wise Rewards Identified by Binary Search**

```
 0.00 Here is a Python code to solve this problem:
 0.00 ```python
+1.00 n = int(input())
+1.00 dangerous_passwords = [input() for _ in range(n)]
+1.00 def is_contained(s, substr):
+1.00     for i in range(len(s) - len(substr) + 1):
+1.00         if s[i:i+len(substr)] == substr:
+1.00             return True
+1.00     return False
-1.00 def generate_strong_password():
-1.00     password = "a"
-1.00     while True:
-1.00         if not any(is_contained(password, dp) for dp in dangerous_passwords):
-1.00             return password
-1.00         password = chr(ord(password) + 1)
-1.00         if password > "z":
-1.00             password = "a"
-1.00 print(generate_strong_password())
 0.00 ```
```

**Line-wise Rewards Predicted by a Learned PRM**

```
-0.08 Here is a Python code to solve this problem:
-0.17 ```python
+0.96 n = int(input())
+0.86 dangerous_passwords = [input() for _ in range(n)]
+0.93 def is_contained(s, substr):
+0.96     for i in range(len(s) - len(substr) + 1):
+0.94         if s[i:i+len(substr)] == substr:
+0.87             return True
+0.87     return False
+0.87 def generate_strong_password():
+0.93     password = "a"
+0.88     while True:
-0.61         if not any(is_contained(password, dp) for dp in dangerous_passwords):
-0.69             return password
-0.66         password = chr(ord(password) + 1)
-0.62         if password > "z":
-0.78             password = "a"
-0.74 print(generate_strong_password())
+0.00 ```
```

Figure 8: Visualization of the learned line-wise rewards. The top gray block displays the problem description, while the bottom section shows a model-generated response with line-wise rewards from different sources. The bottom-left block presents the line-wise rewards identified by binary search, and the bottom-right block presents the line-wise rewards predicted by a learned PRM. The actual reward value is shown at the beginning of each line, and each line is color-coded based on the reward value: lines with rewards closer to -1 are shaded red, while those closer to +1 are shaded green.

