# OpenReview forum: "Process Supervision-Guided Policy Optimization for Code Generation"
_ICLR.cc/2025/Conference — Submitted to ICLR 2025_

### Official Review · Reviewer_2Hjs · 2024-10-31

**Soundness:** 2
**Presentation:** 3
**Contribution:** 2
**Rating:** 3
**Confidence:** 5

**Summary:**

The paper at hand concerns itself with RL fine-tuning of LLMs from unit test feedback, i.e., obtained in code generation tasks by testing the LLM output against a set of unit tests. The main feature here is the use of a process reward model (PRM) to supply dense rewards at multiple points within a generation (token sequence), rather than just at the end (unit test feedback). While PRMs have been used in several domains before, the key contributions are a recipe to obtain a PRM for code as well as ablations on how to best utilize it during RL fine-tuning.

**Strengths:**

With direct experience in the paper's domain I found it, for the most part, easy to read and understand. I liked how the paper ablates different parameters for PRM training and usage. The subject of the paper is also of interest within the community as well as for the deployment of code LLMs in products.

**Weaknesses:**

My main grievance with this paper is that it has very low reproducibility. Models, training data, and evaluation sets are all proprietary; we don't even learn about the size of the models. It does not seem to be a large model, though, indicated by the "lite" name and by the fact that Table 1 only compares to "mini" models from OpenAI and Google. To add to that, a number of crucial details are missing, e.g., hyper-parameters for SFT and RL.

Throughout the paper, the authors claim that their PRM improves exploration. I don't see this claim verified experimentally. Instead, the experiments show better *generalization* to LiveCodeBench and their in-house benchmark, which is not the same. It would be interesting to see if indeed a larger fraction of training problems is solved during training, or whether more diverse solutions are found.

**Questions:**

In Table 1, you compare "Ours-SFT" against "Ours-RL", but the text in L366/367 refers to it as "RL baseline"? Relatedly, since you train the PRM on data produced during the RLHF phase, does that mean "Ours-RL" numbers are the product of two RL training stages? Where is the model after RLHF then (i.e., the "RL baseline")?

---

> ### Author Response · Authors · 2024-11-25
> **Response to Reviewer 2Hjs**
>
> Thank you for your review and for highlighting several important points. We appreciate the opportunity to address your concerns and clarify the aspects you found unclear. Below, we provide detailed responses to each of your comments.
>
> > My main grievance with this paper is that it has very low reproducibility. Models, training data, and evaluation sets are all proprietary; we don't even learn about the size of the models. It does not seem to be a large model, though, indicated by the "lite" name and by the fact that Table 1 only compares to "mini" models from OpenAI and Google. To add to that, a number of crucial details are missing, e.g., hyper-parameters for SFT and RL.
>
> We fully understand your concerns regarding reproducibility. Due to organizational restrictions, we cannot disclose detailed information about our in-house proprietary models and training data. However, to address this issue and improve reproducibility, we have:
>
> - **Statistics on PRM Data for In-House Model Experiments**: We have included additional statistics on the PRM data used in our in-house model experiments in Appendix A of the updated paper. These statistics include token counts, code line distributions.
> - **Training Curves for In-House Model Experiments**: We have attached the training curves for all four settings evaluated in our main experiments in Appendix B. These curves clearly demonstrate that compared to the RL baseline, using PRM for both Dense Rewards and Value Initialization yields the most significant improvements.
> - **Reproduction with Open-Source Models**: To enhance reproducibility, we have reproduced our main results using the open-source model Qwen2.5-7B [1]. The new experiments confirm that our method remains effective with Qwen2.5-7B, further validating its general applicability. Detailed hyperparameters and configurations used in these experiments are provided in Appendix D of the revised paper.
>
> We hope these additional details and open-source experiments address your concerns about reproducibility.
>
> [1] Qwen2.5-7B: https://huggingface.co/Qwen/Qwen2.5-7B
>
> > Throughout the paper, the authors claim that their PRM improves exploration. I don't see this claim verified experimentally. Instead, the experiments show better generalization to LiveCodeBench and their in-house benchmark, which is not the same. It would be interesting to see if indeed a larger fraction of training problems is solved during training, or whether more diverse solutions are found.
>
> Thank you for pointing this out. We have added supporting evidence in the revised paper. Appendix B now includes training curves for all four RL configurations (with/without DenseReward and ValueInit) of our in-house model. These curves demonstrate that models using PRM with DenseReward solve more problems on the training set compared to the RL baseline, indicating improved exploration during training.
>
> We appreciate your suggestion and hope the additional training curves provide clarity on the role of PRM in improving exploration.
>
> > In Table 1, you compare "Ours-SFT" against "Ours-RL", but the text in L366/367 refers to it as "RL baseline"? Relatedly, since you train the PRM on data produced during the RLHF phase, does that mean "Ours-RL" numbers are the product of two RL training stages? Where is the model after RLHF then (i.e., the "RL baseline")?
>
> We apologize for the confusion caused by inconsistent terminology. Here is a clarification:
> - **Terminology**: In Table 1, "RL baseline" refers to the "Ours-RL" configuration where PRM is not used (neither for DenseReward nor ValueInit). This serves as the baseline for comparison.
> - **Two-Stage Training**: The PRM is trained on data collected from the RL baseline checkpoints. After the PRM is trained, we restart RL training using PRM-enhanced rewards (DenseReward and/or ValueInit). This second stage of training produces the results reported for the other three "Ours-RL" configurations in Table 1.
> - **Where the Model is After RLHF**: The RL baseline (or "RLHF model") serves as both a benchmark and the source of checkpoints for PRM data collection. Models trained with PRM restart from scratch, incorporating the PRM rewards into the training process.
>
>
> We hope these updates address your concerns and improve the clarity and impact of our work. Thank you for your feedback, which has been invaluable in refining the paper. If there are additional points you would like us to address, please let us know.

---

> > ### Author Response · Authors · 2024-11-27
> > **Follow-Up Response to Reviewer 2Hjs**
> >
> > Dear Reviewer 2Hjs,
> >
> > We hope our response has addressed all the comments and questions you raised. This is a follow-up response regarding your review.
> >
> > To further clarify your questions about how PRM facilitates more efficient exploration in RL, we have added a new plot (Figure 7) in Appendix B. This plot illustrates the Best-of-K performance of the RL models under all four experimental settings on the training set, complementing the training curves. The rationale is that if a model can solve more unique problems in the training set, its Best-of-K performance should be higher than other models when K is large (i.e., as Best-of-K performance converges).
> >
> > From the plot, we observe that both DenseReward and ValueInit independently improve the Best-of-K performance compared to the baseline. Moreover, when both DenseReward and ValueInit are enabled, the model achieves the highest improvement, with a pass rate increase of nearly 4% at K=30 compared to the baseline (RL without PRM). This demonstrates the significant advantages of PRM in enabling more efficient exploration.
> >
> > As the paper revision period nears its conclusion, we would like to ask if you have any remaining concerns. If so, we would be happy to update our revised paper to address them. We look forward to your feedback.

---

### Official Review · Reviewer_dUPs · 2024-11-01

**Soundness:** 2
**Presentation:** 2
**Contribution:** 2
**Rating:** 3
**Confidence:** 4

**Summary:**

The paper studied using PRM for code generation.

**Strengths:**

First attempt to use PRM for code LLMs.

**Weaknesses:**

The current paper delivery is poor, many critical questions is not clearly explained.

Weakness 1:
The paper discusses using PRM for code generation. But I don't see what specific challenges the proposed method addresses regarding code generation.

The motivation for using PRM in code LLMs (Sec.1 para.2 line 035-043) is that the reward signal is sparse. Sparse reward (or more accurately, the use of bandit feedback) is a common challenge in multi-step reasoning tasks such as math reasoning and code, many research have been conducted to address this challenge, e.g., introducing PRM.
- If the paper limits its scope to code generation, the paper should clearly explain what are the specific challenges it address regarding the code generation task. I notice that unit test feedback is mentioned as a cause for the sparse reward, but I don't consider it as a particular challenge specific in the code generation task. In its essence, test cases serve as a means of verification of the holistic response -- same in its role as evaluating the correctness of a holistic response using the gold-answer in math reasoning. The natural question is: Is previous methods to deal with sparse rewards in math reasoning directly applicable to code generation? If not, the current paper does not clearly explain this.
- If the paper targets at proposing new methods for PRM, the paper should clearly state what is new about the methods.

Based on my current understanding of the paper, the paper answers neither of these.


Weakness 2:
For the current version of this paper, I have doubts on the claimed contributions (line 061)
- From my understanding based on the current paper delivery, the paper only applies previous methods to the code generation task. Reasons detailed as follows:
	- Most importantly, the paper do not clearly explain what additional challenges arise when collecting process supervision for code generation, in contrast to existing works that collect process supervision in reasoning tasks (e.g., OmegaPRM).
	- The paper do not clearly explain how the proposed method differ with previous methods (OmegaPRM), except for the application domain (math versus code).

         OmegaPRM (https://arxiv.org/pdf/2406.06592) introduces the method of automated process supervision using binary search. The paper mentions in line 147: "Instead, we employ an automated approach inspired by techniques used in recent works", but they do not mention how the proposed method differ with OmegaPRM.
- The paper claims to conduct empirical study of "how to properly and effectively integrate PRMs into RL". I see from section 4.3 that, the paper experiment on using PRM as dense rewards, or/and as Value initialization. Using PRM as dense rewards is studied in MathShepherd (https://arxiv.org/abs/2312.08935). The paper draws a conclusion that using PRM to initialize the value function of PPO does not work, but this paper (https://arxiv.org/abs/2406.03816) explores an effective way to employ PRM for value initialization of PPO. So I don't see valuable contributions in this empirical study.
- I question the evaluation of the proposed method. There are no results on common benchmarks for code generations, such as APPS, MBPP, HumanEval

**Questions:**

Better articulate the contributions:
1) What unique challenges arise when applying PRMs to code generation compared to other domains?
2) How does the approach specifically address these challenges? What are the specific differences between the proposed methods and previous methods that inspire this method?
3) The unique challenges in collecting process supervision for code generation versus mathematical reasoning tasks. A point-by-point comparison of their method with OmegaPRM and other relevant approaches, highlighting any novel aspects specific to code generation.
4) Analysis of code LLMs trained with the PRM. Any comparisons on how training with/without PRM improve generated code responses. How does the learned PRM generalize? How does PRM help the value estimate during training?
5) Include results on these specific benchmarks (APPS, MBPP, HumanEval) in their evaluation section. If these benchmarks were not used, provide a clear explanation for why they were omitted and how their chosen benchmarks compare in terms of difficulty and relevance.
Discuss how their results on the chosen benchmarks might translate to performance on these more standard benchmarks.

---

> ### Author Response · Authors · 2024-11-25
> **Response to Reviewer dUPs**
>
> Thank you for your thorough review and for sharing your detailed thoughts and questions. We appreciate the time and effort you invested in reading our paper and providing valuable feedback. We address your concerns point by point below.
>
> > Weakness 1: The paper discusses using PRM for code generation. But I don't see what specific challenges the proposed method addresses regarding code generation.
> >
> > The motivation for using PRM in code LLMs (Sec.1 para.2 line 035-043) is that the reward signal is sparse. Sparse reward (or more accurately, the use of bandit feedback) is a common challenge in multi-step reasoning tasks such as math reasoning and code, many research have been conducted to address this challenge, e.g., introducing PRM.
> >
> > - If the paper limits its scope to code generation, the paper should clearly explain what are the specific challenges it address regarding the code generation task. I notice that unit test feedback is mentioned as a cause for the sparse reward, but I don't consider it as a particular challenge specific in the code generation task. In its essence, test cases serve as a means of verification of the holistic response -- same in its role as evaluating the correctness of a holistic response using the gold-answer in math reasoning. The natural question is: Is previous methods to deal with sparse rewards in math reasoning directly applicable to code generation? If not, the current paper does not clearly explain this.
> >
> > - If the paper targets at proposing new methods for PRM, the paper should clearly state what is new about the methods.
>
> Thank you for recognizing the potential of our approach to generalize to other domains. The simple reason we limited our scope to code generation is that we only conducted experiments in this area and do not wish to claim it can also work for other domains such as mathematical reasoning.
>
> We are aware of existing work on PRMs in the mathematical reasoning domain. Here is our understanding of the difference in the role of process supervision, or PRMs, in each domain:
>
> - In math reasoning, the correctness of the final answer is easy to check, but the correctness of intermediate steps is not. LLMs could output a response with the correct final answer but incorrect reasoning steps. To mitigate this issue, PRMs were introduced to verify intermediate reasoning steps, either to rerank answer responses or to provide guidance signals during decoding.
>
> - In code generation, the correctness of a program is easy to verify with well-designed unit tests. If the program passes all unit tests, there are no intermediate reasoning steps that could be incorrect and need verification. In other words, the code itself serves as a "proof" of solving the problem. Instead, the challenge in code generation is how to efficiently teach LLMs to find a correct "proof" (generate a correct solution) for a problem. In previous work, RLTF introduced the use of unit tests in an online RL framework to let models discover correct "proofs" by themselves. However, a program can only be verified after it is completely generated; thus, the reward from unit tests is sparse and delayed until the end. The role of PRMs here is to provide partial or dense guidance and rewards during the RL process to improve learning efficiency.

---

> > ### Author Response · Authors · 2024-11-25
> > **Continued Response to Reviewer dUPs**
> >
> > > From my understanding based on the current paper delivery, the paper only applies previous methods to the code generation task. Reasons detailed as follows:
> > >    - Most importantly, the paper do not clearly explain what additional challenges arise when collecting process supervision for code generation, in contrast to existing works that collect process supervision in reasoning tasks (e.g., OmegaPRM).
> > >
> > >   - The paper do not clearly explain how the proposed method differ with previous methods (OmegaPRM), except for the application domain (math versus code). OmegaPRM (https://arxiv.org/pdf/2406.06592) introduces the method of automated process supervision using binary search. The paper mentions in line 147: "Instead, we employ an automated approach inspired by techniques used in recent works", but they do not mention how the proposed method differ with OmegaPRM.
> >
> > - Regarding your first point, we did not say or intend to claim that collecting process supervision for code generation is challenging. It simply requires an efficient and practical approach. As a result, we adapted the data collection method from OmegaPRM as our strategy for the entire training pipeline, and we properly cited this paper in Section 3.1.1.
> >
> > - For your second point, we did not claim originality in the data collection method. This paper is not about studying how to collect PRM data for code generation; PRM data collection is just a small part of the overall training process. We adopted the method from OmegaPRM because it is, in our opinion, a relatively efficient and practical approach. Our contribution is to demonstrate how to use this method to properly collect data and train a PRM that can provide a stable process reward signal in an online RL setting.
> >
> > > The paper claims to conduct empirical study of "how to properly and effectively integrate PRMs into RL". I see from section 4.3 that, the paper experiment on using PRM as dense rewards, or/and as Value initialization. Using PRM as dense rewards is studied in MathShepherd (https://arxiv.org/abs/2312.08935). The paper draws a conclusion that using PRM to initialize the value function of PPO does not work, but this paper (https://arxiv.org/abs/2406.03816) explores an effective way to employ PRM for value initialization of PPO. So I don't see valuable contributions in this empirical study.
> >
> > - Regarding your first point about using PRM as a dense reward, we acknowledge that MathShepherd has mentioned this usage in their paper, and we have properly cited and discussed this point in our paper.
> >
> > - For your second point, you state that *"the paper draws a conclusion that using PRM to initialize the value function of PPO **does not** work,"* which is **not** our conclusion. In fact, our paper concludes that when PRM is used both as a dense reward and for value initialization, it yields the best results in our experiments. Regarding the ReST-MCTS* paper (https://arxiv.org/abs/2406.03816) you mentioned, after reviewing it, we did not find any mention of using PPO in their training framework, let alone employing PRM for value initialization of PPO. Their focus is on using PRM as a value model to guide Monte Carlo Tree Search and further improving it through a self-training loop.

---

> > > ### Author Response · Authors · 2024-11-25
> > > **Continued Response to Reviewer dUPs**
> > >
> > > > 1. What unique challenges arise when applying PRMs to code generation compared to other domains?
> > >
> > > As we mentioned above, in code generation, the correctness of a program is easily verified using well-designed unit tests. If a program passes all unit tests, there are no intermediate reasoning steps that could be incorrect and need verification—the code itself serves as a "proof" of solving the problem. Therefore, the challenge in code generation is how to efficiently teach LLMs to find a correct "proof" (i.e., generate a correct solution) for a problem.
> > >
> > > Previous work, such as reinforcement learning from unit test feedback (RLTF), introduced the use of unit tests in an online reinforcement learning framework to enable models to discover correct "proofs" by themselves. However, since a program can only be verified after it has been completely generated, the reward from unit tests is sparse and delayed until the end. The role of PRMs here is to provide partial or dense guidance and rewards during the RL process to improve learning efficiency.
> > >
> > > > 2. How does the approach specifically address these challenges? What are the specific differences between the proposed methods and previous methods that inspire this method?
> > >
> > > The state-of-the-art approach to code generation, RLTF, trains models to generate code that passes all unit tests, enhancing LLMs. However, unit test feedback is sparse, provided only after generating and evaluating entire code snippets, limiting learning efficiency and incremental improvements.
> > > We address this by introducing a PRM into the RL framework. PRM offers dense, line-level feedback on code correctness, enabling more efficient learning and mimicking human-like iterative refinement for improved code generation.
> > >
> > > > 3. The unique challenges in collecting process supervision for code generation versus mathematical reasoning tasks. A point-by-point comparison of their method with OmegaPRM and other relevant approaches, highlighting any novel aspects specific to code generation.
> > >
> > > As mentioned earlier, we do not claim originality for the data collection method. This paper does not focus on studying how to collect PRM data for code generation. Instead, PRM data collection constitutes just one part of the overall training process, and we adopted the method from OmegaPRM, which we consider both efficient and practical. Our contribution lies in demonstrating how to effectively utilize this method to collect data and train a PRM capable of providing a stable process reward signal in an online RL setting.
> > >
> > > > 4. Analysis of code LLMs trained with the PRM. Any comparisons on how training with/without PRM improve generated code responses. How does the learned PRM generalize? How does PRM help the value estimate during training?
> > >
> > > Our experimental results show that using PRM as a DenseReward enables the model to solve more problems compared to the baseline, highlighting PRM's role in facilitating more efficient exploration during RL training. Moreover, when PRM is utilized as both DenseReward and ValueInit, our method achieves the best overall performance. Further details are provided in Appendix B.
> > >
> > > > 5. Include results on these specific benchmarks (APPS, MBPP, HumanEval) in their evaluation section. If these benchmarks were not used, provide a clear explanation for why they were omitted and how their chosen benchmarks compare in terms of difficulty and relevance. Discuss how their results on the chosen benchmarks might translate to performance on these more standard benchmarks.
> > >
> > > We acknowledge your concerns. The primary reason we chose LiveCodeBench is that it is a relatively up-to-date, well-maintained coding benchmark with extensive unit tests for each problem, making it a challenging evaluation, even for models such as GPT-4o, Claude-3.5-Sonnet, and Gemini-Pro-1.5. Additionally, many recent comparisons of LLMs' coding abilities have adopted this benchmark to evaluate "strong reasoning" LLMs, such as DeepSeek R1 and OpenAI O1 (https://api-docs.deepseek.com/news/news1120).
> > >
> > > That said, we understand your concerns and have included evaluation results on HumanEval and MBPP in Appendix C.
> > >
> > >
> > > We appreciate your detailed feedback, which has allowed us to clarify our contributions and refine our presentation. We believe our work offers valuable insights into integrating PRMs for efficient code generation and hope the additional details provided here address your concerns. If there are additional points you would like us to address, please let us know.

---

> > > > ### Comment · Reviewer_dUPs · 2024-11-25
> > > > **Reviewer Response**
> > > >
> > > > Thank you for providing the rebuttal. However, ***my primary concerns regarding the novelty, contribution, and reproducibility of the paper remain unaddressed***. Below, I elaborate on these concerns in detail:
> > > >
> > > > * * * * *
> > > >
> > > > ### **Novelty**
> > > >
> > > > The authors acknowlege that ***the paper primarily applies OmegaPRM to the code generation problem.*** Given that OmegaPRM has already been explored in the context of mathematical reasoning---a domain closely related to code generation---there appear to be no significant methodological innovations. Consequently, the novelty of this work seems limited.
> > > >
> > > > * * * * *
> > > >
> > > > ### **Contribution**
> > > >
> > > > A central concern is whether applying an existing method to a new domain constitutes a substantial contribution. The rebuttal argues that code generation introduces additional challenges compared to mathematical reasoning, particularly in the sparsity and delay of reward signals. However, I find this claim unconvincing.
> > > >
> > > > Sparse rewards due to verification-based reward mechanisms have long been recognized as a challenge in mathematical reasoning (e.g., Wang et al., 2024a). This is acknowledged by the authors themselves, as mentioned in line 359 of the revised manuscript:
> > > >
> > > > > "Our experimental results reveal that using PRM solely as dense rewards significantly outperforms the RL baseline... consistent with findings from Wang et al. (2024a)."
> > > >
> > > > Thus, ***sparse rewards are not unique to code generation; they are a well-known issue in LLM-based reasoning tasks, which have been extensively studied***.
> > > >
> > > > The authors further state in their rebuttal:
> > > >
> > > > > "However, a program can only be verified after it is completely generated; thus, the reward from unit tests is sparse and delayed until the end. The role of PRMs here is to provide partial or dense guidance and rewards during the RL process to improve learning efficiency."
> > > >
> > > > This argument, however, applies equally to mathematical reasoning:
> > > >
> > > > > "A *solution to math queries* can only be verified after it is completely generated; thus, the reward from it is sparse and delayed until the end."
> > > >
> > > > Given this, the assertion that sparse rewards are a uniquely challenging aspect of code generation does not hold. Furthermore:
> > > >
> > > > Regarding my earlier statement:
> > > >
> > > > > "The paper draws a conclusion that using PRM to initialize the value function of PPO does not work."
> > > >
> > > > This statement is based on Table 1, where RL with value initialization shows no improvement over SFT (both yielding an overall score of 28.2). While the authors do not explicitly state this conclusion, ***the results contradict existing research and cast doubt on the significance of the empirical findings.***
> > > >
> > > > Consequently, I believe the contribution of this work is quite limited.
> > > >
> > > > * * * * *
> > > >
> > > > ### **Reproducibility**
> > > >
> > > > The paper does not outline any clear plan for releasing the code or data, which further restricts its reproducibility.

---

> > > > > ### Author Response · Authors · 2024-11-30
> > > > > **Response to Follow-Up Questions from Reviewer dUPs**
> > > > >
> > > > > Thank you for your follow-up questions and concerns regarding our responses. In line with the ICLR 2025 Program Committee's emphasis on fostering meaningful discussions between authors and reviewers, we are happy to engage in a deeper discussion about the novelty, contribution, and reproducibility of the paper. Below, we address these concerns in detail:
> > > > >
> > > > > ---
> > > > >
> > > > > ### **Novelty**
> > > > >
> > > > > We did NOT acknowledge our paper primarily applies OmegaPRM to code generation problem. We acknowledge that our work draws inspiration from OmegaPRM **solely for the PRM data collection** method in our training pipeline. However, **it is incorrect to state that the entire paper focuses on applying OmegaPRM to the code generation domain**.
> > > > >
> > > > > Let us compare OmegaPRM and our work side by side to highlight the distinctions:
> > > > >
> > > > > ---
> > > > >
> > > > > **OmegaPRM**:
> > > > > The contributions of OmegaPRM, as stated in their paper (https://arxiv.org/pdf/2406.06592), are as follows:
> > > > > > - We propose a novel divide-and-conquer style Monte Carlo Tree Search algorithm for automated process supervision data generation.
> > > > > > - The algorithm enables the efficient generation of over 1.5 million process supervision annotations, representing the largest and highest quality dataset of its kind to date. Additionally, the entire process operates without any human annotation, making our method both financially and computationally cost-effective.
> > > > > > - We combine our verifier with weighted self-consistency to further boost the performance of LLM reasoning. We reached 69.4% success rate on the MATH benchmark.
> > > > >
> > > > > From this, it is clear that **OmegaPRM focuses on developing an efficient method for collecting process supervision annotations in the domain of mathematical reasoning**. The core research question addressed in their work is: _How to efficiently obtain a PRM?_
> > > > >
> > > > > ---
> > > > >
> > > > > **Our Work**: The focus of our work is entirely different. **We aim to find a practical recipe for how to use a PRM to improve LLM code generation performance by integrating it into online RL (PPO) training.** Our research is centered on _how can we better integrate PRM into RLTF training paradigm?_
> > > > >
> > > > > Our proposed training pipeline is as follows:
> > > > >
> > > > > RL baseline training -> PRM data generation collection -> PRM training -> RL training w/ PRM
> > > > >
> > > > > In the PRM data generation step, we chose to use binary search to collect process supervision annotations—a subset of the OmegaPRM method without the Monte Carlo Tree Search component. This decision was made because binary search has proven to be both effective and efficient in the domain of mathematical reasoning. Naturally, we adopted this method as the PRM data collection approach in our pipeline.
> > > > >
> > > > > It is important to note that the method used to collect PRM data is interchangeable as long as it provides correct process supervision and can be annotated efficiently. Our work focuses on integrating PRM into RLTF and studying the resulting improvements, rather than innovating the PRM data collection process itself.
> > > > >
> > > > > ---
> > > > >
> > > > > ### **Contribution**
> > > > >
> > > > > Before our work, there has been limited exploration of how to effectively and efficiently use PRMs in online RL (PPO) training. Most prior research on PRMs has focused on training better PRMs as verifiers for reranking model outputs from LLMs (Lightman et al., 2023; Wang et al., 2024a; Jiao et al., 2024; Wang et al., 2024b; Luo et al., 2024). Among these, only Wang et al. (2024a) briefly mentioned the use of PRM as an additional reward in PPO training, supported by a very simple experiment to demonstrate the concept. However, the proper and efficient integration of PRMs into online RL training remains largely unexplored. Our work aims to fill this gap.
> > > > >
> > > > > Our core contribution is a practical recipe for using PRMs to improve LLM code generation performance in PPO training, supported by systematic experiments. These include an empirical study of the optimal PRM data distribution for achieving the best performance and an analysis of the impact of using PRMs as dense rewards and value initialization in PPO training.
> > > > >
> > > > > Key Contributions and Findings:
> > > > > - Demonstrating how dense reward signals can be integrated into the RLTF framework through a PRM.
> > > > > - Presenting a practical recipe for using PRMs to improve LLM code generation performance in PPO training.
> > > > > - Discovering that using PRM for both Dense Reward and Value Initialization in PPO achieves the best results.
> > > > > - Proposing effective methods to mitigate PRM hacking.
> > > > > - Investigating the impact of PRM data distribution on achieving optimal performance.
> > > > > - Highlighting PRMs' capability to enhance code generation in long-horizon scenarios.

---

> > > > > > ### Author Response · Authors · 2024-11-30
> > > > > > **Continued Response to Follow-Up Questions from Reviewer dUPs**
> > > > > >
> > > > > > > This statement is based on Table 1, where RL with value initialization shows no improvement over SFT (both yielding an overall score of 28.2). While the authors do not explicitly state this conclusion, the results contradict existing research and cast doubt on the significance of the empirical findings.
> > > > > >
> > > > > > Regarding your statement, _"This statement is based on Table 1, where RL with value initialization shows no improvement over SFT"_, we would like to correct you: in Table 1, the performance of SFT is 23.5, not 28.2 as you stated. If your intent was to compare the performance of the RL baseline with RL using value initialization, we would like to clarify that the identical performance on LiveCodeBench does **not** imply that RL did not benefit from value initialization. As shown in Figure 7 in Appendix B, when value initialization is used, the learned policy achieves a better Best-of-K performance curve, indicating that it can solve more problems on the training set compared to the RL baseline. Additionally, if you examine the results on HumanEval and MBPP, as you requested, you can see that value initialization improved the RL baseline from 65.1 to 69.8 on HumanEval and from 61.9 to 63.3 on MBPP.
> > > > > >
> > > > > > You also mentioned, _"The results contradict existing research and cast doubt on the significance of the empirical findings"_. To the best of our knowledge, **no existing research has studied the effect of using PRM as the initialization of the value function in PPO training**. If you are referring to the ReST-MCTS* paper (https://arxiv.org/abs/2406.03816) that you mentioned earlier, we have reviewed it carefully. This paper does not discuss the use of PPO in its training framework, nor does it employ PRM for value initialization in PPO. Instead, their focus is on using PRM as a value model to guide MCTS and further refining it through a self-training loop.
> > > > > >
> > > > > > ---
> > > > > >
> > > > > > ### **Reproducibility**
> > > > > >
> > > > > > During the rebuttal period, following the suggestions from all reviewers, we have taken significant steps to address reproducibility concerns by providing additional details and experiments in the Appendix. Below, we outline the efforts we have made:
> > > > > >
> > > > > > 1. **Statistics on PRM Data for In-House Model Experiments**: We have added detailed statistics on the PRM data used in our in-house model experiments in Appendix A of the updated paper. These include token counts, code line distributions, and examples of binary search rewards.
> > > > > > 2. **Training Curves for In-House Model Experiments**: In Appendix B, we have included the training curves for all four experimental settings evaluated in our main experiments. Additionally, we present a plot showing the Best-of-K performance of the RL models under all four settings on the training set.
> > > > > > 3. **Evaluation Results on Open Benchmarks**: As requested in your review, we have further evaluated our models on two additional coding benchmarks: HumanEval and MBPP. The results, provided in Appendix C, are consistent with our main results on LiveCodeBench and InHouseBench, supporting the conclusions presented in the paper.
> > > > > > 4. **Reproduction with Open-Source Models with Details**: To further enhance reproducibility, we have reproduced our main results using the open-source model Qwen2.5-7B. These new experiments confirm that our method is effective with Qwen2.5-7B, further validating its general applicability. Detailed hyperparameters and configurations used in these experiments are included in Appendix D of the revised paper.
> > > > > > 5. **A Typical Example of The Learned Line-wise Rewards**: In Appendix E, we provide a typical example of the line-wise rewards identified by binary search and those predicted by a learned PRM. As suggested by Reviewer ubQx, this example could illustrate key details of our method and offer readers a clearer understanding of its implementation.
> > > > > >
> > > > > > ---
> > > > > >
> > > > > > We appreciate your detailed feedback, which has helped us clarify the novelty, contribution, and reproducibility of our work. We hope these clarifications address your concerns. If you have any additional questions, we would be happy to engage in further discussion. Alternatively, if you feel that we have resolved most of your concerns, we kindly ask you to consider updating your score accordingly.

---

> > > > > > > ### Comment · Reviewer_dUPs · 2024-12-01
> > > > > > > **Reviewer's Final Comments**
> > > > > > >
> > > > > > > The authors indicated there might be potential misunderstandings in my comments.
> > > > > > >
> > > > > > > To ensure my comments are constructive and accountable, I reiterate my previous comments and clarify them as below. I am open to revising my evaluation if the authors can clearly demonstrate inaccuracies in my understanding.
> > > > > > >
> > > > > > > - **Weakness 1: The paper discusses using PRM for code generation. But I don't see what specific challenges the proposed method addresses regarding code generation.**
> > > > > > >
> > > > > > >   The authors stated in their rebuttal
> > > > > > >   > Our Work: The focus of our work is entirely different. We aim to find a practical recipe for how to use a PRM to improve LLM code generation performance by integrating it into online RL (PPO) training. Our research is centered on how can we better integrate PRM into RLTF training paradigm?
> > > > > > >
> > > > > > >   Here are my main points:
> > > > > > >   - ***I disagree that RL from test feedback serves as a unique challenge specific to code generation.*** As detailed in my previous responses, test feedback is essentially a form of verification-based outcome feedback, which has been widely studied in contexts such as mathematical reasoning. The sparse reward issue brought by verification-based outcome feedback has also been studied widely in math reasoning, eliciting popular methods such as PRM.
> > > > > > >   - So my question is: ***in addition to test feedback, what aspects of code generation are truly unique and underexplored when integrating PRM into RL training?***
> > > > > > >
> > > > > > >     In the rebuttal, the authors claimed:
> > > > > > >     > Our core contribution is a practical recipe for using PRMs to improve LLM code generation performance in PPO training, supported by systematic experiments. These include an empirical study of the optimal PRM data distribution for achieving the best performance and an analysis of the impact of using PRMs as dense rewards and value initialization in PPO training.
> > > > > > >
> > > > > > >     However, existing research has already explored some of these aspects. The authors should clarify what is novel in their findings:
> > > > > > >     - MathShepherd (arXiv:2312.08935) studied the use of PRMs as dense rewards.
> > > > > > >     - Rest-MCTS (arXiv:2406.03816) studied PRMs for value initialization in PPO
> > > > > > >
> > > > > > >     Additionally, the above studies are not specifically about unique challenges for code generation, which further underscores my concern.
> > > > > > >
> > > > > > >
> > > > > > > - **Weakness 2: For the current version of this paper, I have doubts on the claimed contributions (line 061)**
> > > > > > >
> > > > > > >   I have concerns regarding the contributions claimed in the paper. Below, I quote the stated contributions and provide my comments:
> > > > > > >
> > > > > > >   - Contribution 1
> > > > > > >   > We propose an effective approach that automatically generates process-level supervision data by identifying the first error line in generated code using binary search. We then train a PRM on this data to generate dense signals during RL training. To the best of our knowledge, we are the first to demonstrate that PRMs can benefit RL from unit test feedback in code generation.
> > > > > > >
> > > > > > >     My comments:
> > > > > > >     ***It is misleading to claim "We propose..." because this is what OmegaPRM have proposed.***
> > > > > > >
> > > > > > >     The authors acknowledged in their rebuttal:
> > > > > > >       > .. we did not claim originality in the data collection method. This paper is not about studying how to collect PRM data for code generation; PRM data collection is just a small part of the overall training process. We adopted the method from OmegaPRM because it is, in our opinion, a relatively efficient and practical approach. Our contribution is to demonstrate how to use this method to properly collect data and train a PRM that can provide a stable process reward signal in an online RL setting.
> > > > > > >
> > > > > > >     If my understanding based on this statement is correct, I suggest the authors replace "We propose..." with "We adopt..." to avoid confusion. And then this contribution appears a bit limited.
> > > > > > >
> > > > > > >   - Contribution 2
> > > > > > >     > We conduct systematic experiments to determine how to properly and effectively integrate PRMs intoRL. Our analysis explores various strategies for training a high-quality code PRM and utilizing PRMs to improve code generation. We summarize our findings into a practical recipe for successfully using PRMs in the context of code generation.
> > > > > > >
> > > > > > >     Similar to my concerns above, ***the authors should clearly outline what is novel about integrating PRMs into RL compared with findings from MathShepherd and Rest-MCTS.***
> > > > > > >
> > > > > > > - **Clarifications**
> > > > > > >
> > > > > > >   The comment regarding "RL with value initialization shows no improvement over SFT (both yielding an overall score of 28.2)" is based on lines 338 and 339, which report the same overall score of 28.2 for both methods. If this is a misconception, please provide clarification.
> > > > > > > ----------
> > > > > > > **Summary**
> > > > > > >
> > > > > > > Based on the above understanding, my overall assessment is
> > > > > > > > **The current paper delivery is poor, many critical questions is not clearly explained.**
> > > > > > >
> > > > > > > Since I cannot properly evaluate the contribution based on the current manuscript, personally I don't think it's ready for publication.

---

> > > > > > > > ### Author Response · Authors · 2024-12-02
> > > > > > > > **Response to Reviewer dUPs's Final Comments**
> > > > > > > >
> > > > > > > > Thank you for your detailed comments. To ensure constructive and meaningful communication, we respectfully request that Reviewer dUPs **carefully review our paper and previous rebuttals**, as most of the concerns raised have already been addressed. Below is our response to the points raised:
> > > > > > > >
> > > > > > > > ---
> > > > > > > >
> > > > > > > > > "Weakness 1: The paper discusses using PRM for code generation. But I don't see what specific challenges the proposed method addresses regarding code generation... in addition to test feedback, what aspects of code generation are truly unique and underexplored when integrating PRM into RL training?"
> > > > > > > >
> > > > > > > > - As mentioned in our paper and rebuttal, the core contribution of our work is a practical recipe for using PRM to improve LLM RL training. **Using PRM to benefit RL training, in itself, is underexplored and non-trivial** for the following reasons:
> > > > > > > >   1. Most PRM-related research focuses on training better PRMs as verifiers for reranking LLM outputs (Lightman et al., 2023; Jiao et al., 2024; Wang et al., 2024b; Luo et al., 2024) and **does not explore their integration into RL training**.
> > > > > > > >   2. Only Wang et al. (2024a) briefly mentions using PRM as an additional reward in PPO training, but this is supported by **very limited experiments**.
> > > > > > > > - Our work is the first attempt to provide an in-depth analysis of how PRMs can benefit LLM RL training. While we acknowledge that code generation and mathematical reasoning share similarities and that our approach can be extended to mathematical tasks, **this does not diminish the significance of our contribution. The main focus of our work is the systematic empirical study of using PRMs to improve LLM RL training, which is both underexplored and non-trivial in the existing literature.**
> > > > > > > >
> > > > > > > > ---
> > > > > > > >
> > > > > > > > > However, existing research has already explored some of these aspects. The authors should clarify what is novel in their findings:
> > > > > > > > > - MathShepherd (arXiv:2312.08935) studied the use of PRMs as dense rewards.
> > > > > > > > > - Rest-MCTS (arXiv:2406.03816) studied PRMs for value initialization in PPO
> > > > > > > >
> > > > > > > > - First, we must clarify (again, as stated in previous rebuttals) that **ReST-MCTS (arXiv:2406.03816) DID NOT study PRMs for value initialization in PPO. We urge Reviewer dUPs to carefully read the ReST-MCTS paper, as this is our third time addressing this comment during the rebuttal period.** To the best of our knowledge, our work is the first to empirically demonstrate that using PRMs for value function initialization in PPO can improve RL performance.
> > > > > > > > - Second, while Math-Shepherd (arXiv:2312.08935) briefly mentions incorporating PRMs as dense rewards in PPO training, their experiments are limited in scope and depth. In contrast, our work provides a comprehensive and systematic investigation into how to effectively integrate PRMs into PPO training. Through our early experiments, we discovered that **simply plugging PRMs into PPO training can easily fail due to reward hacking. Furthermore, without proper PRM data selection and filtering, the signals provided by PRMs can sometimes degrade performance compared to the RL baseline.** In short, it is far from being as straightforward as "plugging PRMs into PPO."
> > > > > > > > - Our research introduces the following critical and novel contributions distilled from our experiments:
> > > > > > > >   1. **Strategies for selecting and filtering PRM training data**: We empirically demonstrate how to optimize PRM data selection to achieve the best performance (Sections 4.2.1 and 4.3).
> > > > > > > >   2. **Empirical methods for mitigating PRM hacking**: We analyze and uncover specific PRM hacking behaviors and propose concrete strategies to mitigate these risks, ensuring stable RL training (Section 4.2.2).
> > > > > > > >   3. **Using PRMs as value function initialization**: We provide the first empirical evidence that PRMs can further enhance RL performance when used for value function initialization, especially when combined with the use of PRMs as dense rewards (Section 4.3 and Appendix B/C).
> > > > > > > >   4. **PRMs can facilitate exploration in RL training**: We show that PRMs enable more efficient exploration during RL training, a crucial advantage for solving complex tasks (Appendix B).
> > > > > > > >
> > > > > > > > These findings are not only novel but are also essential to unlocking the full potential of "embedding PRMs into RL training." We believe they add significant value to the community.
> > > > > > > >
> > > > > > > > ---
> > > > > > > >
> > > > > > > > > "It is misleading to claim 'We propose...' because this is what OmegaPRM has proposed... I suggest the authors replace 'We propose...' with 'We adopt...' to avoid confusion."
> > > > > > > >
> > > > > > > > - We agree with Reviewer dUPs that it would be more appropriate to replace "We propose" with "We adopt," and we will revise the manuscript accordingly.
> > > > > > > > - However, this revision does not diminish the core contribution of our work, which is studying how to effectively integrate PRM into RL training. The data collection method is only a small component of our overall training process.

---

> > > > > > > > > ### Author Response · Authors · 2024-12-02
> > > > > > > > > **Continued Response to Reviewer dUPs's Final Comments**
> > > > > > > > >
> > > > > > > > > > "The comment regarding 'RL with value initialization shows no improvement over SFT (both yielding an overall score of 28.2)' is based on lines 338 and 339, which report the same overall score of 28.2 for both methods. If this is a misconception, please provide clarification."
> > > > > > > > >
> > > > > > > > > - We would greatly appreciate it if Reviewer dUPs could carefully review our paper.
> > > > > > > > >   - Line 337 corresponds to SFT.
> > > > > > > > >   - Line 338 corresponds to RL training without PRM (neither as DenseReward nor ValueInit).
> > > > > > > > >   - Line 339 corresponds to RL training with ValueInit only.
> > > > > > > > > - Therefore, RL with value initialization achieves an improvement of 28.2% - 23.5% = 4.7% compared to SFT.
> > > > > > > > > - Moreover, as shown in Figure 7 in Appendix B, ValueInit leads to a better Best-of-K performance curve, indicating that the learned policy solves more problems on the training set compared to the RL baseline. Additionally, the results on HumanEval and MBPP further support this finding, where ValueInit improves the RL baseline from 65.1 to 69.8 on HumanEval and from 61.9 to 63.3 on MBPP.
> > > > > > > > >
> > > > > > > > > ### **Final Note**
> > > > > > > > >
> > > > > > > > > We hope our response has clarified your concerns. Misunderstandings or misinterpretations can sometimes arise due to the inherent imperfections of communication. However, when both parties are willing to refine their understanding through constructive and meaningful dialogue, it is possible to reach common ground and make meaningful progress.
> > > > > > > > >
> > > > > > > > > You mentioned, _"I am open to revising my evaluation if the authors can clearly demonstrate inaccuracies in my understanding."_ In the responses above, we have addressed multiple factual inaccuracies in your review and comments. We sincerely hope you will reconsider and update your evaluation in light of these clarifications, as it would be greatly appreciated.

---

> ### Comment · Reviewer_dUPs · 2024-12-02
> **Reviewer's Response**
>
> The authors suggest that the reviewer have not carefully read their rebuttal.
>
> However, ***my follow-up responses arise precisely because I have carefully read the rebuttal and found that it does not adequately address my concerns***. My intention in providing these responses is to ensure my reviews are responsible and based on a clear understanding of the manuscript.
>
> I hope the authors recognize that the discussion phase is designed to clarify misunderstandings. ***Instead of assuming reviewers have not read your responses carefully, I encourage the authors to focus on presenting your main ideas more clearly and effectively.***
>
> Returning to the reviews, my aim is to ensure my critiques are not based on misunderstandings of the current manuscript. Based on the provided material, I believe my prior reviews are grounded in a correct interpretation of the work. Importantly, regarding the contributions:
> - contribution 1: The authors have acknowledged that it is misleading to claim, "We propose..." and clarified that it should instead be "We adopt..." This correction validates my point.
> - contribution 2: I want to emphasize that the authors should clearly outline ***in the manuscript*** what is new about integrating PRMs into RL compared with findings from MathShepherd and Rest-MCTS.
>
> To avoid any potential complaints about insufficiently careful reviews, I must stress again: ***These review comments are based on the current manuscript. Since the authors have not revised their manuscript, it remains difficult for readers to properly evaluate the true contributions of the work.***
>
> Regarding the statement, "RL with value initialization shows no improvement over SFT (both yielding an overall score of 28.2)," the confusion arises from the lack of clarity in your presentation. In addition, it is impossible for the readers to infer from the descriptions about Table 1 which line corresponds to SFT and RL, because their are no clear references from the description to the table lines. It is only after the authors clarified that 337 corresponds to SFT and 338 corresponds to RL did this become clear. ***Rather than expecting readers to deduce this, I strongly urge the authors to improve their delivery, e.g., place a clear boundary between SFT and RL, to prevent such ambiguities.***
>
> Additionally, regarding my earlier comment, "The paper draws a conclusion that using PRM to initialize the value function of PPO does not work," this interpretation stems from the statement in line 370: "Interestingly, using PRM solely for value function initialization does not provide notable benefits," which is not followed by any discussion of potential benefits.
>
> Based on the clarifications provided in this thread, I find this claim somewhat misleading. If the authors believe there are indeed benefits to using value function initialization, ***I strongly recommend revising the presentation*** to more accurately reflect these benefits. Otherwise, the readers should have unexpected interpretations.
>
> Overall, the current paper delivery is poor, leading to misconceptions and difficulties in properly in accurately assessing its contributions. My primary concern is the misunderstanding stemming from the paper's current delivery. Determining whether the contributions are sufficient for publication at ICLR seems infeasible, as the lack of substantial revisions leaves no common basis for meaningful discussion.

---

> > ### Author Response · Authors · 2024-12-03
> > **Clarifications and Plans for Improved Presentation**
> >
> > It has been quite a journey discussing our work with Reviewer dUPs, exploring potential areas for improvement and striving to make our submission better from every perspective, including more rigorous evaluations, additional supporting experiments, clarifications of related work, and clearer presentation. To be honest, it is rare to find someone we’ve never met who is willing to provide such detailed and thoughtful feedback on our work. For that, we would like to express our sincere gratitude and appreciation to Reviewer dUPs. Your collaboration has been invaluable in helping us refine and improve our submission to its current level.
> >
> > As we near the conclusion of this discussion, **we have taken time to carefully review the history of our dialogue. Reflecting on our exchanges, we recognize that some of our wording could have been clearer and more amicable. We apologize for any confusion or unintended emotions caused during this process.** Upon reviewing the dialogue, we believe most of the concerns have already been addressed, leaving only a few areas where our presentation could be further refined.
> >
> > We also acknowledge that **a key reason for potential misunderstandings or misinterpretations is the inability to update our manuscript after the revision deadline of _Nov 27th 11:59pm (AoE)_, as set by the ICLR 2025 committee. As a result, any discussions between us and Reviewer dUPs after this date have not been reflected in the current version of the manuscript** (to avoid confusion and for your reference, the current revised version is available at https://openreview.net/revisions?id=Cn5Z0MUPZT). During this rebuttal period, we have assumed that discussions would be based on both the latest manuscript and all rebuttal and follow-up responses. However, Reviewer dUPs stated, "These review comments are based on the current manuscript." We believe this discrepancy might be the source of some of the misunderstandings.
> >
> > That being said, **we fully acknowledge the potential areas for improving the presentation of our work. While we can no longer update the manuscript at this stage, we have outlined the following plans to address these issues** in a potential camera-ready version:
> > - Replace "We propose" with "We adopt" in the contribution section when discussing the PRM data collection method.
> > - Expand the introduction and related work sections to explicitly discuss what is novel about integrating PRMs into RL compared with findings from MathShepherd and Rest-MCTS, as detailed in our rebuttal and follow-up responses.
> > - Add a clear boundary between the results of Our-SFT and Our-RL in Table 1, similar to the style used in Table 4 (Appendix C), to avoid ambiguities.
> > - Refine the statement in line 370, "Interestingly, using PRM solely for value function initialization does not provide notable benefits," to incorporate updated results from HumanEval, MBPP, and the Best-of-K performance presented in Figure 7 of Appendix B.
> >
> > Once again, we sincerely thank you for your efforts in reviewing our work and providing constructive feedback. If you have additional suggestions for improving the presentation of our work, we are open to discussion and happy to adopt them. We genuinely hope that you will reconsider and update your evaluation in light of these clarifications, as it would be greatly appreciated.

---

### Official Review · Reviewer_ubQx · 2024-11-01

**Soundness:** 3
**Presentation:** 3
**Contribution:** 2
**Rating:** 6
**Confidence:** 4

**Summary:**

This paper proposes integrating the Process-Reward Model (PRM) that provides line-level rewards into PPO training for single-turn code generation. The authors design a binary search procedure to gather line-level rewards, which is defined as being able to lead to correct code (the authors use best-of-K sampling to approximate it) for PRM training. The authors propose using the PRM as a source of dense reward and/or value function initialization and evaluating the methods' effect on LiveCodeBench and the in-house benchmark using the in-house model.

**Strengths:**

The contributions of the paper are the following:
1. Adapting the PRM to the PPO pipeline for code generation
2. Details of how the authors design the binary search, collect, and choose PRM training data
3. Empirical results on LiveCodeBench and in-house benchmark show the effectiveness of PRM and analysis to investigate the source of gain

The paper is well-written and easy to follow. It also highlights challenges when incorporating the PRM into the PPO framework (such as reward hacking in Sec 4.3.3). To mitigate reward hacking, it proposes length normalization and neutral labels for comment lines.

**Weaknesses:**

1. My primary concerns are reproducibility, the lack of experimental details, and data transparency, which hinder the community from reproducing the results presented in the work and accessing the needed efforts. This paper uses an in-house model, and part of the results are reported on the in-house benchmark. This is not inherently an issue but means that there needs to be more details in the text/contributions in other components. Therefore, I encourage the authors to include the following:
-----
#### Description of the In-house model
1. What is the parameter size and architecture?
2. What is the size of the SFT set, if available (number of tokens/data points) mentioned in L241 and L337? Could the authors show an example of the SFT data? Could the authors provide some basic configuration of the SFT (e.g., lr, number of gradient steps)?

#### PRM data and training
1. Could the authors give some statistics on the different strategies of PRM training data, if available (number of tokens/data points, avg. lines of code, the distribution of the % or the line number where the line-level label turns -1 from +1)? These statistics could be a valuable add-on to Table 2 and the plain strategy. Could the authors show one example of the reward produced by the binary search?

#### PPO training
1. what is the KL penalty (the value of $\beta$ in Eq(3)) used in the exp.? Some basic hyperparams (lr, steps) would be appreciated.
------
2. The proposed PRM data collection requires model checkpoints from RL baseline training (L252). Therefore, the whole pipeline is: RL baseline training -> PRM data generation collection -> PRM training -> RL training w/ PRM. It makes the first RL baseline training mandatory and increases the computation cost. Did the author explore alternatives, such as using a base policy (e.g., in-house-SFT) to conduct PRM data generation and collection?

**Questions:**

1. On the sensitivity to the $\lambda$: L269 mentions that the authors use different $\lambda$ for correct/incorrect code. I'm curious: Did the author try using the same $\lambda$ or other $\lambda$ during hyperparam sweep? Whether the exp. results deviate much from the ones reported in Table 1?

2. Why the x-axis in Figure 4 could have <1 value as the avg. number of collected response per prompt?

3. L366 says the gain is 5.7% on LiveCodeBench and 12.6% on in-house benchmark relative to the RL baseline. What lines do these numbers correspond to in Table 1? I'm guessing I should be comparing the row of Ours-RL (Dense Reward x, Value Int. x) and the last row. But from Table 1 it's 29.8% - 28.2% = 1.6% and 35.8% - 31.8% = 4%.

---

> ### Author Response · Authors · 2024-11-25
> **Response to Reviewer ubQx**
>
> Thank you for your detailed review and constructive feedback. We appreciate the opportunity to address your concerns and clarify the points raised. Below, we provide detailed responses to your comments and questions.
>
> > My primary concerns are reproducibility, the lack of experimental details, and data transparency, which hinder the community from reproducing the results presented in the work and accessing the needed efforts. This paper uses an in-house model, and part of the results are reported on the in-house benchmark. This is not inherently an issue but means that there needs to be more details in the text/contributions in other components.
>
> We fully understand your concerns regarding reproducibility and appreciate your suggestions for improvement. While restrictions imposed by our organization prevent us from releasing detailed information about the in-house proprietary model and dataset, we have taken significant steps to address this:
>
> - **Statistics on PRM Data for In-House Model Experiments**: We have included additional statistics on the PRM data used in our in-house model experiments in Appendix A of the updated paper. These statistics include token counts, code line distributions.
> - **Training Curves for In-House Model Experiments**: We have attached the training curves for all four settings evaluated in our main experiments in Appendix B. These curves clearly demonstrate that compared to the RL baseline, using PRM for both Dense Rewards and Value Initialization yields the most significant improvements.
> - **Reproduction with Open-Source Models**: To enhance reproducibility, we have reproduced our main results using the open-source model Qwen2.5-7B [1]. The new experiments confirm that our method remains effective with Qwen2.5-7B, further validating its general applicability. Detailed hyperparameters and configurations used in these experiments are provided in Appendix D of the revised paper.
>
> We hope these efforts help bridge the gap in reproducibility and benefit the broader research community.
>
> [1] Qwen2.5-7B: https://huggingface.co/Qwen/Qwen2.5-7B
>
>
> > Description of the In-house model
> > 1. What is the parameter size and architecture?
> 2. What is the size of the SFT set, if available (number of tokens/data points) mentioned in L241 and L337? Could the authors show an example of the SFT data? Could the authors provide some basic configuration of the SFT (e.g., lr, number of gradient steps)?
>
> Due to organizational restrictions, we cannot disclose details about the parameter size, architecture, or specific SFT dataset of the in-house model. However, for the reproduced experiments with Qwen2.5-7B, we provide detailed configurations in Appendix D.
>
> > PRM data and training
> > 1. Could the authors give some statistics on the different strategies of PRM training data, if available (number of tokens/data points, avg. lines of code, the distribution of the % or the line number where the line-level label turns -1 from +1)? These statistics could be a valuable add-on to Table 2 and the plain strategy. Could the authors show one example of the reward produced by the binary search?
>
> Yes, we can definitely share more statistics on the different strategies for selecting PRM training data. Please find these statistics in the Appendix A.
>
> > PPO training
> > 1. what is the KL penalty (the value of $\beta$ in Eq(3)) used in the exp.? Some basic hyperparams (lr, steps) would be appreciated.
> 2. The proposed PRM data collection requires model checkpoints from RL baseline training (L252). Therefore, the whole pipeline is: RL baseline training -> PRM data generation collection -> PRM training -> RL training w/ PRM. It makes the first RL baseline training mandatory and increases the computation cost. Did the author explore alternatives, such as using a base policy (e.g., in-house-SFT) to conduct PRM data generation and collection?
>
> 1. While we cannot disclose proprietary hyperparameters for the in-house model, Appendix D of the revised paper includes the hyperparameters used for Qwen2.5-7B.. We hope this information could also be helpful.
> 2. Yes, your understanding of our proposed training pipeline is correct. It requires first running the RL baseline and using checkpoints sampled during training to collect responses that sufficiently cover the state space. The rationale behind this is that PRM needs to assist RL training throughout the entire process, which necessitates exposure to states that might be visited during training.
>
> In our early experiments, we also included the SFT model in the PRM data collection. However, we observed that this approach negatively impacted the final RL with PRM results, so we decided not to include it. We appreciate your feedback and would like to conduct a more thorough ablation study regarding this during the rebuttal period. However, due to time constraints and resource limitations, we prioritized experiments with the Qwen2.5-7B model and will leave this exploration for future work.

---

> > ### Author Response · Authors · 2024-11-25
> > **Continued Response to Reviewer ubQx**
> >
> > > 1. On the sensitivity to the $\lambda$: L269 mentions that the authors use different $\lambda$ for correct/incorrect code. I'm curious: Did the author try using the same $\lambda$ or other $\lambda$ during hyperparam sweep? Whether the exp. results deviate much from the ones reported in Table 1?
> >
> > In our early experiments, we tested several combinations for $(\lambda\_{\text{correct}}, \lambda\_{\text{incorrect}})$, including $(0.25, 0.25)$, $(0, 0.25)$, and $(0.025, 0.25)$. We found that $(0.025, 0.25)$ provided a small improvement over $(0, 0.25)$ and $(0.25, 0.25)$, so we decided to use $(0.025, 0.25)$ in the final experiments. Overall, the choice of $\lambda$ is not very sensitive as long as it falls within a reasonable range. Specifically, when the code passes all unit tests, less signal from the PRM is needed (so a smaller $\lambda\_{\text{correct}}$ is sufficient). However, when the code fails some unit tests, the PRM needs to provide more fine-grained learning signals (hence, a larger $\lambda\_{\text{incorrect}}$).
> >
> > > 2. Why the x-axis in Figure 4 could have <1 value as the avg. number of collected response per prompt?
> >
> > Apologies for the confusion. The x-axis represents the ratio of the prompts used for PRM data collection to the total prompts in the dataset. A value of <1 indicates that during PRM data collection, we subsampled prompts from the full dataset, resulting in a smaller prompt set.
> >
> > > 3. L366 says the gain is 5.7% on LiveCodeBench and 12.6% on in-house benchmark relative to the RL baseline. What lines do these numbers correspond to in Table 1? I'm guessing I should be comparing the row of Ours-RL (Dense Reward x, Value Int. x) and the last row. But from Table 1 it's 29.8% - 28.2% = 1.6% and 35.8% - 31.8% = 4%.
> >
> > Apologies for the confusion. The gains reported in L366 refer to the relative improvement over the RL baseline. Specifically:
> >
> > (29.8% - 28.2%) / 28.2% = 5.7%
> >
> > (35.8% - 31.8%) / 31.8% = 12.6%
> >
> > We have clarified this in the revised version.
> >
> >
> > We hope these responses address your concerns and provide the necessary clarity. Thank you again for your thoughtful feedback, which has helped us improve the paper significantly. If there are additional points you would like us to address, please let us know.

---

> > > ### Comment · Reviewer_ubQx · 2024-11-25
> > > **Follow-up questions to the Authors**
> > >
> > > Dear Authors,
> > >
> > > Thank you for your response and the improvement you have made during the discussion period. The authors' response did not resolve all my questions regarding the experiment settings, but I think that the authors has made efforts within their "organizational restrictions" to improve the reproducibility. I appreciate a lot the efforts of adding the experiments using a open-weight model.
> > >
> > > That being said, I still have some (most minor) comments that I would appreciate if the authors could clarify, with some of them already in my previous review:
> > >
> > > 1. What is the prompt format used, e.g. [/INST] or chat template, or something else? Does the authors include additional prompt? It would be good if the authors gives the prompt template used.
> > >
> > > 2. Does the model requires to output code only or natural language response around the code is also allowed (e.g. such as the model outputing something like "Here is the solution to your problem: \`\`\`python <code> \`\`\`. This solution solves the problem by ...").  If NL response is allowed, how do the authors label that part with PRM and the training? If NL response is not allowed, from which stage it's code only (SFT, or PRM collection, or RL part)?
> > >
> > > 3. Could the authors show one example of the code solution with reward produced by the binary search / one example of the PRM on the model generated response/code? This could be an example of code produced by the model on LiveCodeBench. This could say a lot of details.
> > >
> > > 4. What is the temperature used in the worker sampling in the RL training? Is it the same as the evaluation temperature 0.2 or it's different?
> > >
> > > 5. I am also interested in the exploration and the increase of diversity, the same question raised by Reviewer 2Hjs. However, I think I need more clarification on this issue: pass@1 is not exactly the same as "unique problem solved" if the authors evaluate using the config in L236-237, i.e., you can have pass@1 increase from 0.1 to 1.0 just because the model changes from solving 1 time out of 10 generations to solving 10 times out of 10 generations, not because of more problems being solved (also I would see this as a drop of diversity if these 10 generations are similar although being correct).
> > >
> > > Nitpick: I think it would be good to clearly state which version of LiveCodeBench (from v1 to v4) this manuscript is benchmarking on, though it could be inferred from the number of problems.

---

> > > > ### Author Response · Authors · 2024-11-27
> > > > **Response to Follow-Up Questions from Reviewer ubQx**
> > > >
> > > > Dear Reviewer ubQx,
> > > >
> > > > We are happy to address your follow-up questions. In the revised paper, we have included additional plots and results to clarify and address your concerns. Below, we provide answers to your questions one by one:
> > > >
> > > > > 1. What is the prompt format used, e.g. [/INST] or chat template, or something else? Does the authors include additional prompt? It would be good if the authors gives the prompt template used.
> > > >
> > > > In our experiments, we restricted LLMs to a single-turn chat completion setting with two roles: "user" and "assistant." The "user" role contains the unmodified problem/prompt from the dataset, while the LLMs generate a response for the "assistant" role. The format is as follows:
> > > > ```
> > > > messages = [
> > > >     {"role": "user", "content": <prompt>},
> > > >     {"role": "assistant", "content": <response>}
> > > > ]
> > > > ```
> > > > The model uses a ChatML-style formatting (https://huggingface.co/docs/transformers/main/en/chat_templating) to structure all information into a generation prompt. No system prompt or few-shot examples were used in our experiments.
> > > >
> > > > > 2. Does the model requires to output code only or natural language response around the code is also allowed (e.g. such as the model outputing something like "Here is the solution to your problem: \```python <code> \```. This solution solves the problem by ..."). If NL response is allowed, how do the authors label that part with PRM and the training? If NL response is not allowed, from which stage it's code only (SFT, or PRM collection, or RL part)?
> > > >
> > > > We did not explicitly restrict the output format of the model during any stage (SFT, PRM collection, or RL). The model could output multiple code blocks (e.g., \```python <code>\```) with natural language before, between, or after the code blocks. To evaluate the generated code, we used a regular expression to extract the content of the first code block as the final generated code.
> > > >
> > > > In cases where the problem specification explicitly instructed the model to output only a \```python <code>\``` block, the model typically produced a single code block without any surrounding natural language.
> > > >
> > > > During PRM data labeling, we assigned a label of 0 to all lines outside of any code block (+1 for correct code lines and -1 for incorrect code lines), treating them the same as comment lines within code blocks (Line 314 in the submission). However, in RL training, we did not differentiate between code lines and natural language lines; PRM was applied uniformly to both types of lines.
> > > >
> > > > > 3. Could the authors show one example of the code solution with reward produced by the binary search / one example of the PRM on the model generated response/code? This could be an example of code produced by the model on LiveCodeBench. This could say a lot of details.
> > > >
> > > > Thank you for bringing this up. In the newly added Appendix E, we included an example problem from the training set along with a response sampled from our model. This is accompanied by the rewards identified through binary search and those predicted by a learned PRM. We hope this example provides the details you are looking for.
> > > >
> > > > > 4. What is the temperature used in the worker sampling in the RL training? Is it the same as the evaluation temperature 0.2 or it's different?
> > > >
> > > > We did not apply any special treatment to the temperature parameter for worker sampling during RL training. Instead, we used a temperature of 1.0 for sample generation.

---

> > > > > ### Author Response · Authors · 2024-11-27
> > > > > **Continued Response to Follow-Up Questions from Reviewer ubQx**
> > > > >
> > > > > > 5. I am also interested in the exploration and the increase of diversity, the same question raised by Reviewer 2Hjs. However, I think I need more clarification on this issue: pass@1 is not exactly the same as "unique problem solved" if the authors evaluate using the config in L236-237, i.e., you can have pass@1 increase from 0.1 to 1.0 just because the model changes from solving 1 time out of 10 generations to solving 10 times out of 10 generations, not because of more problems being solved (also I would see this as a drop of diversity if these 10 generations are similar although being correct).
> > > > >
> > > > > Your concern is entirely valid, and we agree that a higher pass@1 score can result from the model collapsing to a single correct response for a given prompt, rather than solving a broader set of unique problems. To better illustrate how PRM facilitates more efficient exploration in RL, we have added a new plot (Figure 7) in Appendix B. This plot shows the **Best-of-K performance** of the RL models under all four experimental settings **on the training set**, complementing the training curves. The rationale is that if a model can solve more unique problems in the training set, it means that its Best-of-K performance should be higher than other models when K is large (i.e., as Best-of-K performance converges).
> > > > >
> > > > > From the plot, we observe that both DenseReward and ValueInit independently improve the Best-of-K performance compared to the baseline. Furthermore, when both DenseReward and ValueInit are enabled, the model achieves the highest improvement, with a pass rate increase of nearly 4% at K=30 compared to the baseline (RL without PRM). This demonstrates the significant advantages of PRM in enabling more efficient exploration.
> > > > >
> > > > > > Nitpick: I think it would be good to clearly state which version of LiveCodeBench (from v1 to v4) this manuscript is benchmarking on, though it could be inferred from the number of problems.
> > > > >
> > > > > Apologies for the confusion. We used LiveCodeBench v3, which includes problems released between May 2023 and July 2024, totaling 612 problems. We have also clarified this in the revised version of the manuscript.
> > > > >
> > > > > We hope that the additional information and clarifications provided in this response address your concerns. We appreciate your thoughtful feedback, which has been instrumental in improving the clarity and reproducibility of our work. If you have any further questions or need additional clarification, please don’t hesitate to reach out.

---

> > > > > > ### Comment · Reviewer_ubQx · 2024-11-28
> > > > > > **Thank you for your response**
> > > > > >
> > > > > > Dear Authors,
> > > > > >
> > > > > > Thank you for clarifying my follow-up questions. I'm raising my score to reflect the improvement made to the submission during the discussion period.

---

### Official Review · Reviewer_Sfwe · 2024-11-02

**Soundness:** 4
**Presentation:** 3
**Contribution:** 3
**Rating:** 8
**Confidence:** 3

**Summary:**

The paper introduces a practical approach for training models with process-oriented feedback for code generation. The approach uses a novel automatic (LLM + unit test -based) data generation process to create a dataset of code with quality labels on every line. The line-by-line labels can be used to trained a Process Reward Model (PRM), which can in turn be used as a dense reward signal for training code models with RL. The approach is shown to outperform the baseline RL-trained code models which are trained on sparse rewards from unit tests. The authors perform careful experiments and ablations to motivate each part of their design.

**Strengths:**

Nice work overall! I enjoyed reading this and have become more bullish on PRMs as a general research direction after this.
- Very readable and easy to understand.
- Clearly a lot of work was put into it: Devises a novel practical approach for training models with PRMs for code generation, with careful work put into ablations and studying each component.
- Competitive with the SOTA RLTF approach
- Clever method to generate process-level supervision data to train PRMs! The idea of using a best-of-K oracle to label whether each code prefix is feasible or contains unrecoverable errors is a non-obvious but effective way to generate data.
- Section 4.2 and 4.3 were great! Well-reasoned experiments and execution, with great experiment-backed insights on how best to use PRMs in this domain.
  - Particularly appreciated the detail-to-attention uncovering PRM Hacking and implementing mitigations.
  - I like that the main results (Table 1) independently shows the effect of introducing [Value Init], [Dense Reward], and [Value Init + Dense Reward]! Very clear.

**Weaknesses:**

No clear weaknesses come to mind. There were some choices made which I had different ideas about, but this is not a critique of the work or the claims, so I have put those into the Questions section.

**Questions:**

- > To ensure that the PRM training data effectively covers the state space the language model may encounter during the next RL training phase, we sample policy models from various stages of the RL baseline training. Specifically, we select 4 checkpoints evenly spaced throughout the RL baseline model’s training process. For each checkpoint, we sample n responses for each coding prompt in the training dataset Dtrain. For each sampled response, we apply the binary search labeling procedure described in Algorithm 1

  For the oracle, is this also using the policy model checkpoints? I see "Our method leverages the model’s own capabilities to generate completions for partial code prefixes and uses automated testing to assess their correctness" which suggests that the policy model itself is used to generate the best-of-K samples. I can appreciate that not requiring a separate oracle model is nice because it is self-contained, but I think this will result in a suboptimal dataset compared to e.g. using a fully-trained RLTF code model as the oracle.

- Reading Section 4.2.2 on RL Training makes me think: Wouldn't a different scheme of data labelling, which labels each line with the "marginal contribution of the line toward success" be more effective than simply rating [0: infeasible, 1: feasible]? Specifically, my intuition is that each added line should improve the success rate of the oracle given K attempts, so my proposed reward is something like "the reward for step M should be the `(success_rate_of_oracle_at_step_M - success_rate_of_oracle_at_step_Mminus1)`". This captures the idea that each line should increase the likelihood of the program succeeding, and naturally avoids reward hacking by simply adding more lines. Of course, this is easier said than done, I expect the process to be noisy, but this formulation for the dataset seems to be better aligned to the true objective.

- (Not a weakness, just a typo) In Section 3.2, I think the authors meant to do citep instead of just cite: "In mathematical domains, LLMs may generate correct answers with faulty reasoning Lightman et al. (2023), making intermediate verification essential" and "While preliminary attempts have been made to incorporate PRMs into RL trainingWang et al. (2024a)"

---

> ### Author Response · Authors · 2024-11-25
> **Response to Reviewer Sfwe**
>
> Thank you for your detailed review and thoughtful feedback. We appreciate your positive comments on the novelty, clarity, and thoroughness of our work, as well as your insightful questions and suggestions. Below, we address your queries and observations.
>
> >For the oracle, is this also using the policy model checkpoints? I see "Our method leverages the model’s own capabilities to generate completions for partial code prefixes and uses automated testing to assess their correctness" which suggests that the policy model itself is used to generate the best-of-K samples. I can appreciate that not requiring a separate oracle model is nice because it is self-contained, but I think this will result in a suboptimal dataset compared to e.g. using a fully-trained RLTF code model as the oracle.
>
> Yes, you are correct. In our experiments, we use the policy model checkpoints as the oracle to generate the best-of-K samples.This self-contained approach simplifies the pipeline and avoids dependency on external models. However, we agree that employing a fully-trained RLTF code model as the oracle could potentially improve the quality of the dataset by providing more robust labels.
>
> While we did not perform an explicit ablation study on this aspect, related work has observed that for larger values of K, an RL policy may not consistently outperform an SFT policy in terms of best-of-K performance [1]. This suggests that further exploration of alternative oracle designs, such as a fully-trained RLTF model, could yield interesting insights and improvements. We appreciate your suggestion and will consider it in future work.
>
>
> [1] Wang, E., Cassano, F., Wu, C., Bai, Y., Song, W., Nath, V., ... & Zhang, H. (2024). Planning in natural language improves LLM search for code generation. arXiv preprint arXiv:2409.03733.
>
>
> > Reading Section 4.2.2 on RL Training makes me think: Wouldn't a different scheme of data labelling, which labels each line with the "marginal contribution of the line toward success" be more effective than simply rating [0: infeasible, 1: feasible]? Specifically, my intuition is that each added line should improve the success rate of the oracle given K attempts, so my proposed reward is something like "the reward for step M should be the (success_rate_of_oracle_at_step_M - success_rate_of_oracle_at_step_Mminus1)". This captures the idea that each line should increase the likelihood of the program succeeding, and naturally avoids reward hacking by simply adding more lines. Of course, this is easier said than done, I expect the process to be noisy, but this formulation for the dataset seems to be better aligned to the true objective.
>
>
> Thank you for this insightful suggestion. Estimating the marginal contribution of each line to the overall success rate is indeed a compelling idea, as it could provide a more fine-grained and objective measure of progress. Implementing this scheme would require estimating the value function for each line under the oracle and computing the difference, (e.g.,$ V(\text{line}\_M) - V(\text{line}\_{M-1}) $).
>
> However, there are practical trade-offs to consider. Using the Monte Carlo method to estimate the value function with K rollouts would require $O(N^2 \times K) $ token generations for a trajectory of length $N$.
> In contrast, our current binary search labeling procedure reduces this too $ O(N \log N \times K) $, significantly lowering computational cost. Given the expense of LLM generation, we prioritized the more computationally efficient approach while achieving robust performance.
>
> > (Not a weakness, just a typo) In Section 3.2, I think the authors meant to do citep instead of just cite: "In mathematical domains, LLMs may generate correct answers with faulty reasoning Lightman et al. (2023), making intermediate verification essential" and "While preliminary attempts have been made to incorporate PRMs into RL trainingWang et al. (2024a)"
>
> Thank you for catching this error. We have corrected it in the revised version to ensure proper citation formatting.
>
> We greatly appreciate your thoughtful review and the opportunity to improve our work based on your suggestions. If there are additional points you would like us to address, please let us know.

---

### Meta-Review · Area_Chair_SWU9 · 2024-12-22

**Metareview:**

> The paper at hand concerns itself with RL fine-tuning of LLMs from unit test feedback, i.e., obtained in code generation tasks by testing the LLM output against a set of unit tests. The main feature here is the use of a process reward model (PRM) to supply dense rewards at multiple points within a generation (token sequence), rather than just at the end (unit test feedback). While PRMs have been used in several domains before, the key contributions are a recipe to obtain a PRM for code as well as ablations on how to best utilize it during RL fine-tuning.

This looks like it's good work, but the paper is problematic. It's not clear from the paper alone what are the differences with OmegaPRM, and this a notable part of the contribution. Moreover, the main problem is that the paper is not easy to compare to (let alone reproduce). During rebuttal, the authors worked on applying their RL+PRM method on Qwen 2.5 7B and LiveCodeBench, but before that it was just a proprietary (in house) model on a proprietary (in house) dataset. The two scores of "3" were made by honest reviews, and while those reviewers didn't change their score in the light of the rebuttal, I do not think it would have been likely they both change them to 6+, even with the changes provided by the authors.

I suggest to rework the paper to put the focus on the algorithmic contribution with at least a comparable setup, or better yet a reproducible one, so that the paper gets read from the get go on its merits, not its (major) flaws.

**Additional Comments On Reviewer Discussion:**

Rebuttal changed the paper significantly but not enough to change its fate.

---

### Decision · Program_Chairs · 2025-01-22

Reject